# 🔴Mars: Situated Inductive Reasoning in an Open-World Environment

**Xiaojuan Tang** [1,3]
xiaojuan@stu.pku.edu.cn

**Jiaqi Li** [3]
lijiaqi@bigai.ai

**Yitao Liang** [1,3]
yitaol@pku.edu.cn

**Song-chun Zhu** [1,2,3]
sczhu@bigai.ai

**Muhan Zhang** [1,3,*]
muhan@pku.edu.cn

**Zilong Zheng** [3,*]
zlzheng@bigai.ai

[1] Institute for Artificial Intelligence, Peking University
[2] Department of Automation, Tsinghua University
[3] State Key Laboratory of General Artificial Intelligence, BIGAI

**https://marscrafter.github.io**

## Abstract

Large Language Models (LLMs) trained on massive corpora have shown remarkable success in knowledge-intensive tasks. Yet, most of them rely on pre-stored knowledge. Inducing new general knowledge from a specific environment and performing reasoning with the acquired knowledge—*situated inductive reasoning*, is crucial and challenging for machine intelligence. In this paper, we design Mars, an interactive environment devised for situated inductive reasoning. It introduces counter-commonsense game mechanics by modifying terrain, survival setting and task dependency while adhering to certain principles. In Mars, agents need to actively interact with their surroundings, derive useful rules and perform decision-making tasks in specific contexts. We conduct experiments on various RL-based and LLM-based methods, finding that they all struggle on this challenging situated inductive reasoning benchmark. Furthermore, we explore *Induction from Reflection*, where we instruct agents to perform inductive reasoning from history trajectory. The superior performance underscores the importance of inductive reasoning in Mars. Through Mars, we aim to galvanize advancements in situated inductive reasoning and set the stage for developing the next generation of AI systems that can reason in an adaptive and context-sensitive way.

## 1 Introduction

Imagine a scenario: in the United States, you drive on the right side of the road. When you travel to the UK, you might initially find it strange how people drive. However, you soon realize that driving on the left is the norm here and adapt yourself to the new rule. Inductive reasoning, a capacity that identifies underlying rules, mechanisms, or general claims of *unobserved* experience based on past *observations*, undoubtedly plays a pivot role in scientific discoveries as well as in the conduct of our everyday affairs. Research on the origin and justifications of such inductive aptitude can date back to the 1900s. David Hume, one of the most influential philosophers in human nature, presented a critical dilemma as follows:

---

*Corresponding authors

38th Conference on Neural Information Processing Systems (NeurIPS 2024) Track on Datasets and Benchmarks.

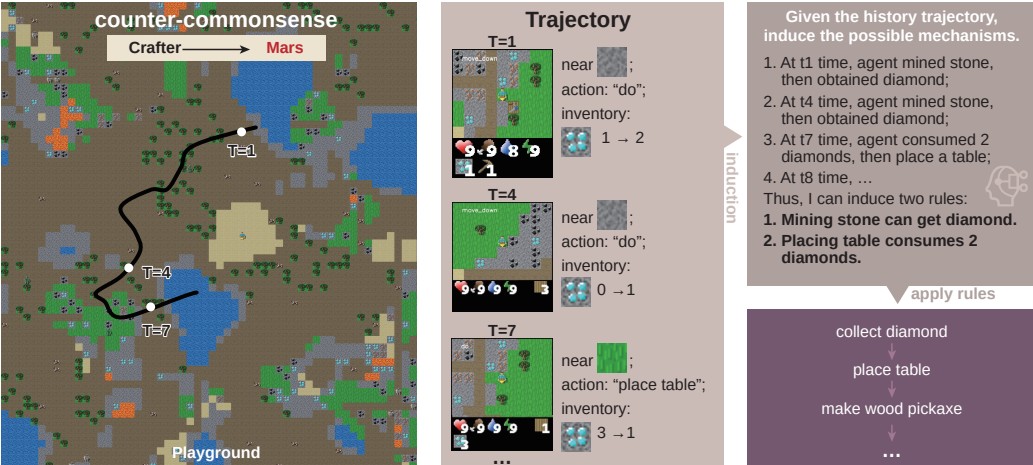

Figure 1: 🔴 **Mars**, an open-world environment for situated inductive reasoning, involves inductive reasoning through active interaction and applying newly acquired rules to make context-sensitive decisions. First, built on Crafter, we introduce counter-commonsense elements to design Mars. Agents interact with the environment and accumulate historical trajectories. For example, an agent might observe that regardless of time or location, mining stone always yields diamonds; using 2 diamonds can craft a table. Consequently, the agent can induce rules "Mining stone yields diamond" and "Placing table consumes 2 diamonds". When tasked with making a wooden pickaxe, the agent can apply these rules to plan and execute specific actions in different contexts.

> "Why from this (present) experience we form any conclusion *beyond* those past instances, of which we have had experience."
>
> — Hume [1896], *A Treatise of Human Nature*

Hume's words, also known as "The Problem of Induction", imply two fundamental questions of inductive reasoning: ❶ How to summarize and form conclusions from the *present*, and live observations? ❷ Based on summarizations, how to derive *inductive* conclusions (*i.e.*, rules or general claims) beyond past experiences? To answer these two questions, we anticipate two crucial aspects existing in the process of inductive reasoning.

- **Situatedness:** Question ❶ poses a challenge to understand situations dynamically and reason with the present knowledge accordingly, *i.e.*, situated reasoning. Cognitive studies also indicate that cognition cannot be separated from context and that learning occurs in a situated activity that encompasses social, cultural, and physical contexts [Brown et al., 1989, Roth and Jornet, 2013, Greeno, 1998, Lave and Wenger, 1991].
- **Abstractiveness:** The capability of summarizing observations into abstract "conclusions" that go beyond old experiences, *e.g.*, symbols, logics, rules and causal relations, is highlighted in question ❷. Prior research works on inductive reasoning [Zhang et al., 2021a, Raven, 2003, Nye et al., 2020] mostly focus on this side by formalizing such a process within rigorous logical forms and performing evaluations directly based on inductive logical rules.

To cover both aspects, we introduce 🔴 **Mars**, a novel interactive environment that aims at benchmarking models' capabilities on **situated inductive reasoning**, in which models are required to quickly derive new general knowledge (rules) from interactions within a specific environment and apply the newly acquired knowledge effectively in a new context, rather than merely storing, retrieving or using pre-existing knowledge. Here, "Mars" is not meant to represent the actual planet Mars. Instead, it symbolically represents a "Martian" environment with knowledge and conditions that differ from commonsense (or "Earth" knowledge). Built on the foundation of Crafter [Hafner, 2021]), an open-world survival game, we modify three categories of the default game mechanisms: terrain, survival settings, and task dependencies (§2). Sampling from the combinations of the three kinds of mechanism changes, Mars can generate numerous different worlds with distinct properties. In each world, agents need to continuously interact with the environment and accomplish tasks un-

Table 1: Comparison between Mars and related benchmark.

| Datasets | task | type | interactive? | situated? | induction? | evidence | source |
|---|---|---|---|---|---|---|---|
| ARC [2019] | q.a. | visual | ✗ | ✗ | ✓ | pre-defined | synthetic |
| MiniSCAN [2020] | q.a. | visual | ✗ | ✗ | ✓ | pre-defined | synthetic |
| ACRE [2021a] | q.a. | visual | ✗ | ✗ | ✓ | pre-defined | synthetic |
| List Functions [2020, 2021b, 2022] | q.a. | symbol | ✗ | ✗ | ✓ | pre-defined | human-written |
| RAVEN [2019] | q.a. | visual | ✗ | ✗ | ✓ | pre-defined | synthetic |
| DERR [2022] | q.a. | language | ✗ | ✗ | ✓ | pre-defined | Wikipedia |
| bAbI-16 [2015] | q.a. | language | ✗ | ✗ | ✓ | pre-defined | synthetic |
| STAR [2024] | q.a. | visual | ✗ | ✓ | ✗ | - | human activity videos |
| SQA-3D [2022] | q.a. | 3D | ✗ | ✓ | ✗ | - | 3D indoor |
| SOK-Bench [2024d] | q.a. | visual | ✗ | ✓ | ✗ | - | real-world activities |
| IQA [2018] | q.a. | visual | ✓ | ✗ | ✗ | - | indoor |
| MP3D-EQA [2019] | q.a. | 3D | ✓ | ✗ | ✗ | - | indoor |
| 🔴 Mars (Ours) | policy | visual[1] | ✓ | ✓ | ✓ | open-ended | synthetic |

til the end of their lifespan. However, they cannot merely leverage their prior knowledge (such as "consuming cows increases health") since these pre-stored "earth" knowledge might no longer apply on Mars. Instead, they have to actively induce the rules of the new world, which provides a valuable testbed for their situated inductive reasoning abilities.

In §2.3, strict principles govern the design of each sampled new world. These principles ensure resource balance, supply exceeding demand, and the achievability of each task. By adhering to these guidelines, Mars avoids creating a purely fantastical or unstable world, allowing the agents to effectively utilize their extensive prior knowledge.

Our work is closely related to the recent surge of LLM-as-agents [Brown et al., 2020, Zhang et al., 2022, Chowdhery et al., 2023], where LLMs behave as reasoning agents and present impressive capabilities in embodied planning and acting, question answering, machine translation, *etc*. [Ahn et al., 2022, Du et al., 2023, Wang et al., 2024a, Shinn et al., 2023, Bubeck et al., 2023, Gao et al., 2023, Wang et al., 2023a, Mihaylov et al., 2018, Wang et al., 2024b, Zhang et al., 2024, Cai et al., 2023a,b, Lin et al., 2023, Wang et al., 2024c, Cai et al.]. However, most of these tasks are rich in world knowledge, allowing LLMs to exploit their vast stored knowledge to perform the tasks instead of reasoning. Recently, some research conduct counter-commonsense experiments through QA tasks [Wu et al., 2023, Saparov and He, 2022, Dasgupta et al., 2022, Tang et al., 2023, Han et al., 2022]. They primarily evaluate model's ability to apply some given knowledge (rules) to reason in new context without learning new rules from the given context. Another line of inductive reasoning work [Mirchandani et al., 2023, Kim et al., 2022, Weston et al., 2015, Yang et al., 2022] provides pre-defined evidence (input-output pairs) and evaluates performance on some new input, instead of actively interacting with the environment to collect evidence, inducing new rules, and applying the induced rules in context. Comparisons with relevant tasks and benchmarks are listed in Table 1.

In §3, we carefully select seven representative worlds with varying difficulty (deviation from commonsense) from our proposed Mars. We then evaluate them using state-of-the-art online reinforcement learning methods and LLM agents. Moreover, inspired by the prior success of reflexion [Shinn et al., 2023], we propose a novel LLM-based pipeline, *induction from reflection* (IfR), where LLM is forced to engage in a reflective thinking process to induce new game rules. Our findings indicate that current models perform poorly in these settings, highlighting the need for improved situated inductive reasoning skills that go beyond static knowledge application.

## 2 The 🔴 Mars Environment

Mars is designed as an interactive open-world survival game, aiming at evaluating an agent's situated inductive reasoning capability, as depicted in Figure 1. Building on the foundation of Crafter [Hafner, 2021], Mars can strategically alter certain commonsense, including terrain, survival settings and task dependencies, while adhering to certain principles related to resource balance, item quantities, and task achievability.

### 2.1 Basic Setting: Crafter

Crafter [Hafner, 2021] is an open-world survival game designed to evaluate a wide range of general abilities, including robust generalization, deep exploration and long-horizon reasoning. In this de-

---
[1]We also provide the interface to translate visual information into language.

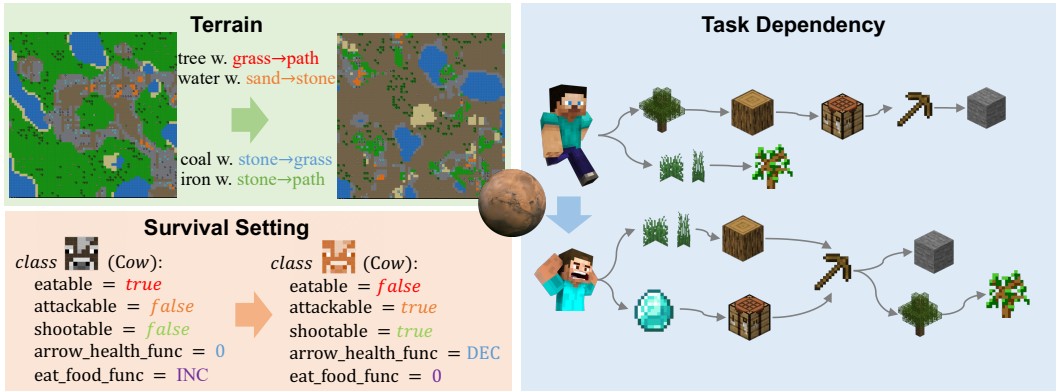

Figure 2: Examples of three kinds of modification to commonsense elements. Please refer to Appendix A.5 for more details.

manding environment, the agent (*e.g.*, a policy model) is asked to unlock all achievements while ensuring its survival. Each episode generates a unique world featuring diverse terrains such as grasslands, lakes, and mountains, randomly populated with entities like cows, trees, and zombies. The game world is structured on a $64 \times 64$ grid, yet the agent's observation is restricted to a $7 \times 9$ grid, with an additional $2 \times 9$ grid space for displaying inventory and status, making Crafter a partially observed environment. At each step, the agent gathers information about the surrounding terrain, its health, food, drink, energy levels, and inventory. Following this, the agent must select an action from a set of 17 possible actions.

## 2.2 Modification: From Crafter to Mars

To challenge the agent with an environment that deviates from prior (parametric) knowledge and necessitates situated inductive reasoning, we introduce targeted modifications to typical commonsense elements, classified into three categories (Figure 2):

**Terrain** Terrain includes two aspects: terrain distribution and terrain effect. In the default Crafter setting, common terrain distributions are predictably arranged, *e.g.*, minerals like coal, iron, and diamonds are discovered near stone formations. Terrain effects involve whether a terrain can be traversed and whether doing so benefits or harms the agents health, or even results in death. These terrain characteristics guide the agents exploration strategies and efficiency. We disrupt these norms by altering the distribution and effects of these elements, *i.e.*, trees may now grow near sand rather than grass and lava is not hot.

**Survival Settings** We introduce a novel axis of variation in survival dynamics. It mainly involves characteristics of entities like cows, zombies, skeletons, plants (edibility, aggressiveness, proximity effects, mobility) as well as the impact on the agents status level (health, food and drink) when consuming these entities and drink. For example, in Crafter world, cows can enhance the agents food levels upon consumption; in this altered reality, cows may exhibit hostile behaviors.

**Task Dependency** Agents can collect many resources by mining some materials and use them to build tools and place objects. To this end, we classify them into three kinds of achievements: collecting, placing and crafting. Please refer to Appendix A.5 for more details.

*Collecting* Collecting involves using a tool to mine items and obtain resources while leaving behind some terrain materials. Modifications include altering resources to visually resemble something else, leading to unexpected outcomes (*e.g.*, coal appearing as stone so that mining stone will collect coal instead). Tools for mining are randomly selected (hand, wooden, iron, stone pickaxe), and the leftover materials are randomly sampled. Liquid terrains (water, lava, sand) may leave behind creatures (*e.g.*, zombies) with default behaviors.

*Placing and Craftering* Modifications to placing focus on the ignitability of materials, which is randomized for wood, stone, coal, iron, and diamond. Crafting tables can be made from any material while furnaces, which are used for smelting, must be crafted from non-flammable substances. Regarding crafting achievements, we assume that the names of items often reflect their materials. Thus, we do not alter the raw materials used for tools. Instead, we consider whether a table or fur-

nace is required based on the ignitability of the materials. For items that are ignitable, both a table and a furnace are required, whereas for non-flammable items, a table suffices.

## 2.3 Principles of new world

While we can sample numerous new worlds following the above procedure, we carefully designed several strict principles so that they are not completely fantastical and are always playable.

- The new world does not introduce additional resources or objects; it only modifies the functions or effects of existing game objects and materials. To ensure playability, we guarantee that each collected item has at least one obtainable method and each tool has a practical use, motivating the agent to engage in crafting. We maintain the same achievements as the default Crafter environment to allow for fair comparisons in subsequent experimental evaluations.
- We adhere to the resource balance principle. For every resource that can be increased by some event, there must be a corresponding event that can decrease the resource, maintaining a balance. For instance, if the agent loses health when attacked by a cow, there should be scenarios where the health level increases, such as eating zombie.
- We also ensure that each achievement is achievable. For example, if mining wood requires a wooden pickaxe, but crafting a wooden pickaxe requires wood, this creates a deadlock. To prevent such scenarios, we construct an and-or tree and use the depth-first search (DFS) algorithm to verify that each task in the technology tree has a viable path to the root node, confirming the feasibility of each task. Additionally, we also develop an automated program to evaluate terrain distribution, walkable materials, and task dependencies generated by item recipes, ensuring all items are accessible. For example, assuming that coal and stone are not directly traversable, if we place diamonds around the stone (because mining stone is a precondition for mining diamonds based on task dependency and diamonds are not walkable), the agent is unable to reach the stone and complete the "mine stone" task.
- We ensure supply exceeds demand: the quantity of items required for task achievements must be greater than what the world provides. For instance, if wood requires collecting at least five diamonds, but the world does not has enough diamonds. Additionally, our world includes mechanisms for renewable resources, such as mining a tree potentially leaving behind a coal terrain. This dynamic aspect means that the availability of resources cannot be measured statically. To address this, we develop an algorithm that simulates the process of unlocking all achievements within the Tech Tree to test whether the dynamically regenerating resources of the world are sufficient to complete all tasks.

## 3 Evaluation on Mars

### 3.1 Evaluation Setup

**Metrics** We use three evaluation metrics as in Hafner [2021] to assess the performance of models situated inductive reasoning abilities: i) The **reward** metric reflects the agents skills. Each time an agent unlocks an achievement, the reward increases by 1. When an agents health increases or decreases by 1, the reward adjusts by +0.1 or -0.1, respectively. ii) The **success rate** is defined as the proportion of achievements unlocked during the episodes. iii) The **overall score** averages the success rate of the 22 achievements in log-space (to account for differences in their difficulties) as: $S = \exp(\frac{1}{N} \sum_{i=1}^{N} \ln(1 + s_i)) - 1$.

**Evaluation worlds** In Mars, we meticulously select seven different worlds, focusing on individual modifications to terrain, survival settings, and task dependency: Terrain, Survival, and Task Dep. respectively; we concurrently modify two types of commonsense rules: Terr. Surv., Terr. Task., and Surv. Task.; as well as all three types simultaneously: All three. We also conduct experiments in the Crafter setting (*i.e.*, Default). Configurations of worlds are in Appendix M.

### 3.2 Baselines

To evaluate Mars, we design (1) RL-based methods: PPO [Schulman et al., 2017], DreamerV3 [Hafner et al., 2023]; (2) LLM-based methods: ReAct [Yao et al., 2022], Reflexion [Shinn et al., 2023], revised framework motivated by skill library [Xin et al., 2023, Wang et al., 2023a] and (3) our proposed framework induction from reflection. Note that RL-based methods individually train a model for each world with 1 million training steps. They do not truly solve the problem of *quickly adapting to new environments* in situated inductive reasoning scenarios. Here, we conduct

the experiments only to provide the reference. To assess the situated inductive reasoning capabilities of RL-based methods, we also further test different worlds using the DreamerV3 trained in Crafter (Appendix D). Our primary comparison focuses on the LLM-based in-context learning methods. See Appendix B for explanations of why LLMs can be used to evaluate the ability to perform situated inductive reasoning.

**RL-based methods**: **PPO** takes images as input and learns to output actions through policy gradient descent. In our implementation, we use a convolutional neural network (CNN) to parameterize the policy gradient. We use stable_baselines3 [Raffin et al., 2021] to conduct the experiment with the default parameters. **DreamerV3** [Hafner et al., 2024] is a general and scalable algorithm based on world models using fixed hyperparameters with 3 neural networks. It succeeds across domains by accommodating different signal magnitudes and balance terms in their objectives for various domains. We adopt the default parameters provided in the source code[2]. All agents are trained for 1 million environment steps with reward and tested over 20 independent trials.

**LLM-based methods**: Considering that LLMs cannot accept image inputs, we provide a wrapper that gives text descriptions of gameplay screen, including the coordinates of objects, agent's status and inventory (Appendix A.4). **ReAct** [Yao et al., 2022] interleaves the generation of reasoning traces and task-specific actions. **Reflexion** [Shinn et al., 2023] builds on top of ReAct by incorporating self-reflection, allowing the model to reflect on past experiences. When the historical trajectory exceeds a certain token limit (set to 3896 tokens here), the model is provided with the reward and score in its context for reflective thinking. Based on JARVIS-1 and Voyager [Wang et al., 2023a,b], we further simplify the framework to adapt to Mars, called **Skill Library** (Appendix C).

## 3.3 Induction from Reflection (IfR)

Building on the **Skill Library** framework, we further introduce the *induction from reflection* module in *controller*, as depicted in Figure 3. When the *controller* finishes a subgoal (including "succeed", "failed" or "timeout"), we force LLM to engage in reflective thinking to induce possible game mechanisms based on the agents historical trajectory. The derived rules are then stored in a *rule library*, which the task proposer, planner, and controller can use.

For Skill Library and IfR, we set the learning episodes to 5. For ReAct and Reflexion, which rely on in-context memory instead of external memory, we restrict them to use a finite context window (10 steps or 3896 tokens trajectory). For all LLM-based methods, we use the GPT-4-0125-preview model [Achiam et al., 2023] through OpenAIs API, with a temperature of 0.7. Other hyper-parameters (*e.g.*, top_k) are kept at their default settings. The full prompts for all different methods are provided in Appendix L.

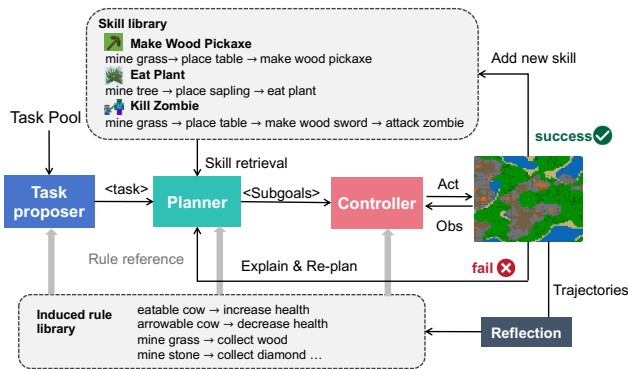

Figure 3: **An illustration of the Induction from reflection pipeline for Mars.** Given the selected task and the agent's observation, *planner* decomposes the task into a sequence of subgoals. *Controller* then outputs specific actions to accomplish these subgoals. Successful plans are stored in the *skill library*, while failed plans prompt the agent to perform self-explanation and replan. *Rule library* is updated through reflection on the controller's execution. By performing inductive reasoning, it saves possible game rules for proposer, planner, and controller using.

## 3.4 Main Results

Table 2 presents the performance of various methods across different environments. Notably, all baseline models exhibit a performance decline when transitioning from the Default to Mars scenarios, with the extent of the decline dependent on the type (*e.g.*, terrain, survival, and task dependency)

---

[2]https://github.com/NM512/dreamerv3-torch

Table 2: **Performance comparison of RL-based and LLM-based methods.** Results for LM models are summarized over 9 independent trials while RL methods over 20 independent trials. $\pm$ captures standard deviations. The best results are in red while the seconds are in blue.

| Metrics | Mod. Type | RL-based methods | | LLM-based methods | | | |
|---|---|---|---|---|---|---|---|
| | | PPO | DreamerV3 | ReAct | Reflexion | Skill Library | Ours |
| **Reward** | Default | $1.9^{\pm1.4}$ | $11.5^{\pm1.6}$ | $7.7^{\pm1.6}$ | $6.0^{\pm1.7}$ | $8.0^{\pm2.1}$ | $9.0^{\pm2.3}$ |
| | Terrain | $-0.1^{\pm0.6}$ | $9.3^{\pm2.2}$ | $7.4^{\pm2.7}$ | $6.4^{\pm3.0}$ | $9.5^{\pm2.9}$ | $8.0^{\pm3.7}$ |
| | Survival | $-0.6^{\pm0.5}$ | $8.6^{\pm4.1}$ | $6.4^{\pm3.7}$ | $4.6^{\pm3.9}$ | $7.9^{\pm2.9}$ | $7.7^{\pm3.7}$ |
| | Task. Dep | $2.1^{\pm1.2}$ | $8.8^{\pm2.8}$ | $5.0^{\pm2.1}$ | $3.2^{\pm1.6}$ | $1.5^{\pm1.9}$ | $5.6^{\pm2.9}$ |
| | Terr. Surv. | $0.0^{\pm0.7}$ | $7.1^{\pm2.1}$ | $6.7^{\pm2.5}$ | $4.9^{\pm2.5}$ | $3.0^{\pm2.5}$ | $6.8^{\pm1.9}$ |
| | Terr. Task. | $-0.7^{\pm0.3}$ | $6.6^{\pm0.7}$ | $4.8^{\pm2.0}$ | $5.3^{\pm2.5}$ | $5.5^{\pm1.5}$ | $6.9^{\pm1.8}$ |
| | Surv. Task. | $-0.6^{\pm0.4}$ | $9.6^{\pm3.4}$ | $1.5^{\pm1.3}$ | $1.0^{\pm1.6}$ | $2.3^{\pm1.5}$ | $3.3^{\pm1.4}$ |
| | All three. | $0.1^{\pm0.8}$ | $5.1^{\pm1.8}$ | $0.7^{\pm1.6}$ | $-0.4^{\pm0.7}$ | $-0.5^{\pm0.5}$ | $0.1^{\pm0.5}$ |
| **Score (%)** | Default | $1.3^{\pm1.7}$ | $14.2^{\pm1.3}$ | $8.0^{\pm1.5}$ | $5.3^{\pm0.9}$ | $8.3^{\pm1.3}$ | $13.0^{\pm2.1}$ |
| | Terrain | $0.3^{\pm0.1}$ | $13.0^{\pm1.6}$ | $7.6^{\pm2.6}$ | $7.4^{\pm1.6}$ | $11.9^{\pm3.4}$ | $11.8^{\pm2.9}$ |
| | Survival | $0.2^{\pm0.0}$ | $10.8^{\pm2.8}$ | $8.0^{\pm0.6}$ | $5.5^{\pm1.7}$ | $9.7^{\pm2.0}$ | $11.0^{\pm3.7}$ |
| | Task. Dep | $1.7^{\pm0.6}$ | $12.1^{\pm1.9}$ | $4.6^{\pm1.6}$ | $2.2^{\pm0.8}$ | $1.5^{\pm0.6}$ | $6.9^{\pm2.5}$ |
| | Terr. Surv. | $0.4^{\pm0.1}$ | $7.9^{\pm1.3}$ | $7.1^{\pm3.0}$ | $4.7^{\pm1.6}$ | $2.8^{\pm0.6}$ | $6.7^{\pm0.8}$ |
| | Terr. Task. | $0.1^{\pm0.1}$ | $4.2^{\pm0.1}$ | $3.8^{\pm0.3}$ | $5.5^{\pm1.7}$ | $4.1^{\pm0.7}$ | $7.1^{\pm2.5}$ |
| | Surv. Task | $0.1^{\pm0.1}$ | $15.9^{\pm2.6}$ | $1.3^{\pm0.2}$ | $1.1^{\pm0.1}$ | $1.9^{\pm0.1}$ | $2.1^{\pm0.4}$ |
| | All three. | $0.6^{\pm0.2}$ | $4.0^{\pm0.3}$ | $1.0^{\pm0.3}$ | $0.2^{\pm0.1}$ | $0.2^{\pm0.0}$ | $0.6^{\pm0.0}$ |

and the number of modifications. This underscores that **Mars presents significant challenges for current methodologies**. Although our proposed method shows some improvement, its suboptimal performance in the "All three" modified world highlights the urgent need for further research in this complex reasoning context.

For RL-based methods, DreamerV3 outperforms most LLM-based methods, likely due to its extensive exploration, having been trained for 1 million steps. However, in the "All three." scenario, DreamerV3 achieves only a 4% score. This suggests that **counter-commonsense modifications introduce additional complexity to the game mechanics**, thereby increasing the learning difficulty for RL-based models and hindering rapid adaptation.

For LLM-based methods, we observe that altering terrain and survival settings has minimal negative impact on the Skill Library model. However, **changing task dependencies significantly degrades performance**. This is particularly evident when the visual appearance of resources is modified (*e.g.*, mining stone yields wood)—under the "Task Dep." setting, the Skill Library achieves a reward of 1.5 compared to ReAct's 5.0. This likely occurs because ReAct's step-by-step reasoning is more adaptable than the Skill Library's multi-step planning approach. Additionally, the Skill Library's memory only retains *successful* subgoal sequences, making it challenging to accurately assess the real mechanisms for task completion. Consequently, this leads to incorrect plans and erroneous exploration paths (Appendix G).

This issue also motivates us to introduce "induction from reflection" in LLM-based controller module. It encourages the controller to reflect on the counter-commonsense situations and further explore the actual game mechanisms. From the results, we observe that models equipped with the induction capabilities outperform those without, highlighting the importance of inductive reasoning in a counter-commonsense environment.

## 3.5 Further Analysis

We further plot the success rate of unlocking achievements by the Skill Library model, comparing the default world (Crafter) to the "Task. Dep" world in Mars, as shown in Figure 4. Most achievements involving task dependency category (*e.g.*, collecting, placing) experience a significant drop in performance. Even tasks related to survival, such as collecting drinks, are slightly affected. The performance for "kill something" tasks is likely impacted due to the difficulty in making a sword. Interestingly, the unlock rate for the "collect diamond" task in the "Task. Dep" world is higher than in the "Default" world. This is because, in the modified world, diamonds can be directly mined

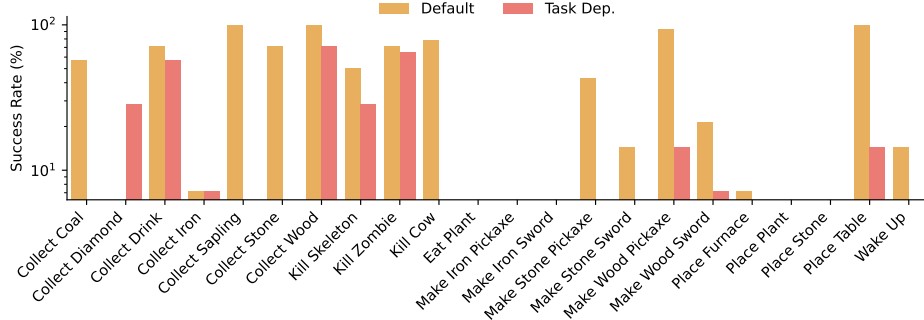

Figure 4: Success rate of unlocking 22 different achievements in log scale by Skill Library model.

by hand, making it a straightforward, one-step process that is easy to discover through exploration. However, for the more complex two-step task, "place table", which requires using two diamonds, the performance is still poorer. These results again highlight that Mars is challenging for current methods. Next, we conduct experimental analyses on situated reasoning and inductive reasoning separately. More details and case studies are presented in Appendix I.

**Situated reasoning:** We evaluate the situated reasoning abilities of ReAct by providing it with game rules of each world in context. As shown in Line 2 and Line 4 of Table 3, LLMs perform better when provided with necessary rules. However, "Surv. Task. w/ rules" has lower scores than "Default w/ rules", indicating significant challenges in understanding and applying counter-commonsense rules. This observation aligns with findings from previous works [Dasgupta et al., 2022, Tang et al., 2023, Saparov and He, 2022].

Table 3: Results of ReAct when provided with game rules.

| Mod. Type | Score | Reward |
|---|---|---|
| Default | 8.0% | 7.7 |
| Default w/ rules | 11.6% | 7.9 |
| Surv. Task. | 1.3% | 1.5 |
| Surv. Task. w/ rules | 9.2% | 4.9 |

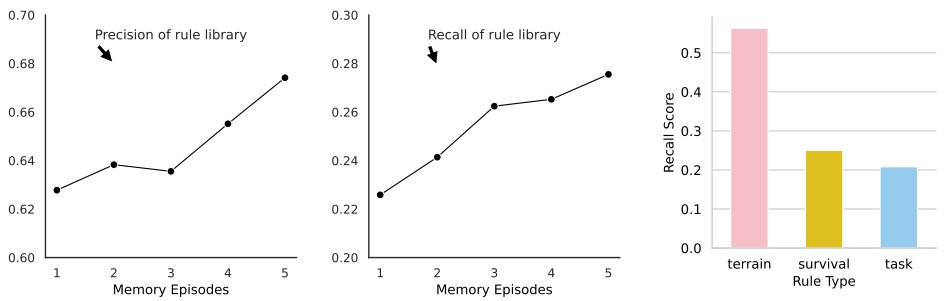

Figure 5: Evaluation of rule library

**Inductive reasoning:** We further evaluate the benefits of IfR. For induced rules (stored in the rule library) and the ground truth rules (provided in the world configurations) in natural language, we measure the precision of the predicted rules and the recall of the ground truth rules using GPT-4 as an evaluator. The results, shown in Figure 5, indicate that the scores improve as the rule library grows with increased memory episodes. However, the recall score of about 28% indicates that there is still much room for improvement. When analyzing the rule types, it can be found that terrain rules are the easiest to induce, followed by survival setting rules, and finally task dependency rules. The results align with the observations in Table 2—modifying task dependency leads to poorer performance compared to terrain and survival settings, likely due to a larger induction search space.

# 4 Related Work

**Inductive Reasoning.** Inductive reasoning is the ability to infer general principles from specific observations or evidence and apply them to novel situations, which is fundamental to human intelligence [Peirce, 1868]. A few researchers have proposed a myriad of tasks to evaluate inductive reasoning in AI. Representative benchmarks include vision-based reasoning [Mirchandani et al., 2023, Kim et al., 2022, Xu et al., 2023a, Moskvichev et al., 2023, Zhang et al., 2021a, 2019, Barrett et al., 2018, Webb et al., 2020, Hill et al., 2019, Raven, 2003][3], program-based induction [Rule, 2020, Zhang et al., 2021b, Srivastava et al., 2022], natural language-based [Weston et al., 2015, Yang et al., 2022] and sequence-to-sequence tasks [Nye et al., 2020]. These tasks ususally consist of 2-5 input-output pairs and a test problem. The goal is to infer the rule (*e.g.*, transformation, function) from given examples and apply them to the problem input. Simultaneously, some studies evaluate inductive reasoning capabilities of pretrained large LMs [Gendron et al., 2023, Tang et al., 2023, Xu et al., 2023b, Han et al., 2024, Xu et al., 2023a, Alet et al., 2021]. Honovich et al. [2022] infer an underlying task from a few demonstrations. Wang et al. [2023c], Qiu et al. [2023] proposes hypothesis search and iterative refinement to improve inductive reasoning abilities.

**Situated Reasoning.** Situated reasoning requires agents to understand the situation and surroundings from a dynamic view, then reasoning and accomplishing complex tasks accordingly. SQA3D [Ma et al., 2022] focuses on situated question answering in 3D scenes, requiring agents to comprehend and localize their position and orientation. STAR [Wu et al., 2024] requires agents understand and abstract the dynamic situations presented in the videos. SOK-Bench [Wang et al., 2024d] emphasizes understanding and applying both situated and general knowledge for problem-solving. Other works in embodied question answering place agents in interactive environments, such as MP3D-R2R [Anderson et al., 2018], MP3D-EQA [Wijmans et al., 2019], IQA [Gordon et al., 2018], and EmbodiedQA [Das et al., 2018]. These benchmarks and datasets typically rely on **factual knowledge** (which is only specific to the current situation) extracted from surroundings or some pre-existing commonsense knowledge to perform deductive reasoning accordingly. However, Mars introduces counter-commonsense game mechanisms, which not only require a deep understanding of the current situation but also necessitate learning **general rules** through inductive reasoning.

# 5 Conclusion

In this paper, we introduce ⬤**Mars**, designed to evaluate models' situated inductive reasoning abilities in adaptive and context-sensitive way. Key components, including terrain, survival settings, and task dependencies, are modified according to certain principles. In Mars, agents are required to actively interact with their surroundings, learn to derive new general knowledge, and perform reasoning using the acquired knowledge. Furthermore, we propose *Induction from Reflection* method, which compels LLMs to perform inductive reasoning from historical trajectories. This approach has demonstrated better performance compared to other LLM-based methods, underscoring the significance of inductive reasoning in counter-commonsense environments.

**Limitations and Future Work** Despite the improved performance of IfR compared to other LLM-based method, the overall performance remains suboptimal. In addition to the model's limitations in identifying the underlying causes of observations, this could be due to the limited exploration time provided by the five episodes and the relatively inefficient exploration process. Future research could focus on enhancing the models exploration efficiency and utilizing induced rules to make more informed guesses. For example, if an agent discovers that lava is walkable and safe, it might hypothesize that water could be dangerous due to resource balance. Additionally, future models could be designed to *automatically* identify the causes and perform inductive reasoning when encountering a new environment, eliminating the need for enforced induction from historical trajectories.

# 6 Acknowledgements

I express my gratitude to my advisors for their guidance and to my peers for their valuable suggestions. This work is partially supported by the National Key R&D Program of China (2022ZD0160300), the National Natural Science Foundation of China (62376031).

---

[3]Note that they can also be represented in text format to evaluate LLMs.

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

# Appendix

## A  Additional Mars details

### A.1  Benchmark URLs and Links

Mars is published under the open-source MIT license on Github https://github.com/XiaojuanTang/Mars. Code for all the benchmark models are available within the same GitHub repository. We provide detailed descriptions at https://github.com/XiaojuanTang/Mars/blob/master/README.md.

The documentation covers:

- Step-by-step instructions for setting up the Mars environment.
- Guidelines on how to load and use various world configurations.
- Descriptions of the configurations. See details in Appendix A.5 and Appendix M. Our code repository also includes a demo video for each world to enhance understanding of these configurations.
- Benchmark code and examples of how to run the benchmarks.

### A.2  Implementation details

The detailed hyperparameters of the RL-based baselines are provided in Section 3.2. Specifically, for the PPO experiment, we utilize the default parameters from the stable_baselines3 library, while for DreamerV3, we adopt the default parameters as specified in the source code (available at https://github.com/NM512/dreamerv3-torch). All agents are trained for 1 million environment steps with rewards and tested over 20 independent trials. For further details on the LLM baselines, please refer to Section 3.2. Additional prompts can be found in Appendix L. We also provide the code including all baselines, at https://github.com/XiaojuanTang/Mars, which can help with reproducing our results.

### A.3  Maintenance and Long Term Preservation

The Mars dataset is an interactive environment built on the Crafter framework, designed to evaluate situated inductive reasoning in agents. The authors of Mars are committed to maintaining and preserving this environment. Ongoing maintenance also encompasses tracking and resolving issues identified by the broader community after release. User feedback will be closely monitored via the GitHub issue tracker.

### A.4  Details of environment descriptor

The gameplay screen consists of a $9 \times 9$ grid $((i,j)|1 \le i,j \le 9)$. The top seven rows provide a local view of the world; each cell $(i,j)$ is associated with a predefined background (*e.g.*, "grass", "stone", "sand") and potentially an object (*e.g.*, "tree", "cow"). The bottom two rows represent the agent's status (*e.g.*, "health", "food") and item inventories, which include images of items (e.g., "stone", "stone sword") and the quantity of each item in the inventory.

Our environment descriptor processes the gameplay screen as input and outputs a textual description of the screen. This description includes the agent's action, nearby block information, agent status, and inventory details. Specifically:

- **Action**: The descriptor outputs the specific action taken by the agent, such as "I took action move_left".
- **Nearby Block Information**: For cells containing objects, the descriptor focuses on the objects; for cells without objects, it focuses on the background. It first identifies all types of backgrounds and objects within the $7 \times 9$ grid. The text descriptor outlines the background material closest to the agent and enumerates all objects, including their coordinates. For example, "I see: (objects with coordinate) path is in front of me. <path(-1, 0), path(1, 0), path(0, -1), path(0, 1), path(-1, -1), path(1, -1), path(-1, 1), path(1, 1), stone(-2, -1), tree(-3, 0)>".

- **Agent Status**: The descriptor provides the agent's health, food, drink, and energy levels, each of which ranges from 0 to 9.
- **Inventory**: The descriptor outputs the types and quantities of items present in the inventory.

Each frame typically generates a text description with a token length ranging from 120-180 tokens. To ensure the LLM's context window is not overwhelmed, we save the historical trajectory within a certain token limit.

Below is a comprehensive example:

> **Agent's observation:**
> I took action move_left.
> I am on the path.
> I see: (object with coordinate)
> tree is in front of me.
> <tree(-1, 0), path(1, 0), stone(0, -1), path(0, 1), stone(-1, -1), path(1, -1), stone(-1, 1), path(1, 1), water(-3, 0), sand(-3, 3)>
> My status: <health: 9/9, food: 9/9, drink: 9/9, energy: 9/9>
> I have nothing in your inventory.

### A.5 Details of modified commonsense elements

In this section, we introduce the modified commonsense elements in detail, including terrain, survival settings and task dependency. We also provide the configuration of Crafter world. The configurations of Mars world are in Appendix M.

#### A.5.1 Terrain

Modification of terrain involves two aspects: terrain distribution and terrain effect. The terrain material includes water, grass, stone, path, sand, tree, lava, coal, iron and diamond,

- **Terrain Distribution:** In the default Crafter environment, common terrain distributions are predictably arranged: sand typically encircles bodies of water; trees are prevalent near grasslands; and minerals like coal, iron, diamonds, and lava are found near stone formations. The player is usually born in grass. In Mars, we modify the terrain neighbors or swap terrain names to change the terrain distribution. Specifically, for the first modification type, we sample the surroundings of coal, iron, diamond, lava, tree, player, and water terrains with one of the terrain materials. For example, coal could be placed near grasslands. Note that we ensure each type of terrain material is sampled, and each item is accessible. For the second modification type, we exchange different terrain names. For instance, we swap the positions of stone and iron terrains.

- **Terrain Effect:** This involves whether a terrain can be traversed and whether doing so benefits or harms the agent's health or even results in death. To this end, we assign each terrain material (except trees, due to their inherent height, despite the 2D game's limitations) three attributes: walkable, walk_health, and dieable. We randomly assign values to these three attributes: walkable: [True, False]; walk_health: [-1,0,+1]; dieable: [True, False]. For example, envision a planet where you discover energy stones unlike anything on Earth, or where, surprisingly, lava is not hot. Note that if the terrain material is not walkable, the dieable and walk_health attributes have no practical significance.

Here is the Crafter setting:

Terrain distribution of Crafter:

```
terrain_neighbour:
    coal: stone
    iron: stone
    diamond: stone
    lava: stone
    tree: grass
```

```
    player: grass
    water: sand
```

Terrain effect of Crafter:

```
terrain_effect:
    stone: {walkable: false, walk_health: 0, dieable: false}
    diamond: {walkable: false, walk_health: 0, dieable: false}
    coal: {walkable: false, walk_health: 0, dieable: false}
    iron: {walkable: false, walk_health: 0, dieable: false}
    water: {walkable: false, walk_health: 0, dieable: false}
    lava: {walkable: true, walk_health: 0, dieable: true}
    grass: {walkable: true, walk_health: 0, dieable: false}
    path: {walkable: true, walk_health: 0, dieable: false}
    sand: {walkable: true, walk_health: 0, dieable: false}
    tree: {walkable: false, walk_health: 0, dieable: false}
```

### A.5.2   Survival settings

This modification mainly involves the characteristics of objects, including cows, zombies, skeletons, ripe-plants, as well as drinks like water and lava. For example, in Crafter world, cows can enhance the agents food levels upon consumption; zombies approach and harming the agent; skeletons shoot arrows that cause damage to the agent; water replenishes the agents drink level. In this altered reality, cows may exhibit hostile behaviors, consuming a ripe plant could increase hunger due to its digestion-enhancing properties, and consuming overly salty zombie flesh could increase thirst (if the zombie is edible in this world). Specifically, for objects, we set the following attributes:

- eatable: Indicates if the object is edible;
- eat_health_damage_func: The impact on the agent's health when consumed (increase, decrease, or no effect);
- inc_food_func: The impact on the agent's food level when consumed.
- inc_thirst_func: The impact on the agent's thirst level when consumed;
- arrowable: Indicates if the object can perform shooting actions;
- arrow_damage_func: the impact on the agent's health when shot.
- closable: Indicates if the object will move towards the agent;
- can_walk: Indicates if the object can move.
- closable_health_damage_func: The impact on the agent's health when the object is near.

For drinks, we set the following attributes:

- inc_drink_func: The impact on the agent's drink level when consumed.
- inc_health_func: The impact on the agent's health level when consumed.
- inc_food_func: The impact on the agent's food level when consumed.

We randomly assign the value to those attributes to modify the survival setting. For example, zombies shooting arrows that cause damage to the agent, *i.e.*, "arrowable=True, arrow_damage_func=-1"; drink lava can increase agent's health, *i.e.*, "inc_health_func=+1".

The survival setting of Crafter is as below:

```
cow:
    eatable: true
    arrowable: false
    closable: false
    can_walk: true
    closable_health_damage_func: 0
    eat_health_damage_func: 0
    arrow_damage_func: 0
```

```
    inc_food_func: 1
    inc_thirst_func: 0
zombie:
    eatable: false
    arrowable: false
    closable: true
    can_walk: true
    closable_health_damage_func: -1
    eat_health_damage_func: 0
    arrow_damage_func: 0
    inc_food_func: 0
    inc_thirst_func: 0
skeleton:
    eatable: false
    arrowable: true
    closable: false
    can_walk: true
    closable_health_damage_func: 0
    eat_health_damage_func: 0
    arrow_damage_func: -1
    inc_food_func: 0
    inc_thirst_func: 0
plant:
    eatable: true
    arrowable: false
    closable: false
    can_walk: false
    closable_health_damage_func: 0
    eat_health_damage_func: 0
    arrow_damage_func: 0
    inc_food_func: 1
    inc_thirst_func: 0
```

### A.5.3 Task Dependency

Agents can collect many resources, such as saplings, wood, stone, coal, iron and diamond and use them to build tools or place objects. Many of the resources require tools that require even more basic tools and resources, leading to a technology tree with several levels. Typically, agents start by collecting wood, crafting a wooden pickaxe, then progressing to stone, coal, and so on, with diamond collection being the ultimate and most challenging achievement. However, in our new environment, these dependencies are disrupted; for example, collecting diamonds no longer requires an iron pickaxe, and collecting wood now requires specific tools. To this end, we consider three kinds of achievements: collecting, placing and crafting. Refer to Appendix A.5 for more details.

**Collecting:** The task of collecting involves mining a terrain material with a tool or hand to receive items while leaving other materials behind. For example, chopping down a tree by hand may yield wood while leaving grass. Following this, we implement three different changes to received items:

- *Visual Misleading*: In this modified world, mining a resource may yield an unexpected item. For instance, what appears to be coal could actually yield stone instead, as stone may visually resemble coal in this unconventional world. Specifically, we randomly permute the expected items (including wood, stone, coal, iron, diamonds and sapling) for terrains (including grass, trees, stone, coal, iron, and diamonds). For liquid terrains such as water and lava, the output (*e.g.*, whether agents receive a drink) is randomly assigned as "True" or "False". This approach selectively disrupts the visual alignment of solid materials without confusing them with liquids, maintaining the challenge of non-common knowledge rather than creating a completely fantastical or symbolic world.

- *Traditional Association with Exceptions*: Contrary to the first, this easier modification maintains the traditional association between an items appearance and its material composition, i.e., mining

stone yields stone. However, trees, while still visually resembling trees, can produce unconventional items such as diamonds or coal. Similarly, mining grass can also yield stone. To achieve this, for stone, coal, iron and diamonds, mine them still yield stone, coal, iron and diamonds respectively. For tree and grass, we random sample from items {wood, stone, coal, iron, diamonds and sapling} and ensure each item has at least one obtainable method.

- *Probabilistic Outcomes*: Building on the second modification, we introduce a probabilistic element where mining a resource might yield multiple potential outputs with certain probabilities. For instance, mining stone with a wooden pickaxe might primarily produce stone but also offer a chance (e.g., 10% probability) of finding coal. This probabilistic approach, where resource extraction can be unpredictable and yield secondary resources, increases the games difficulty while also simulating real-world scenarios. Specifically, for stone, coal, iron, and diamond, mining them not only yields their respective items but also has a 10% probability of dropping other items, including wood, stone, coal, iron, and saplings, which are randomly sampled.

In addition to changes in received items, we also modify the tools used for mining. These tools are randomly sampled from {null (using hands), sapling, wooden pickaxe, stone pickaxe, and iron pickaxe}. Each tool must have a practical use to motivate the agent to engage in crafting. After mining, the material left behind is also randomly sampled from different terrain types. For instance, mining a tree may leave behind another tree, indicating that trees in this world grow rapidly and are inexhaustible. For liquid-like terrain such as water, lava and sand, there may even be a chance of leaving behind creatures like zombies, cows, or skeletons, each behaving according to their default characteristics.

Here is one example of modified collecting tasks:

```
collect:
    tree: {require: {iron_pickaxe: 1}, receive: {coal: 1}, leaves:
     {material: iron, object: null}}
    stone: {require: {}, receive: {stone: 1}, leaves: {material:
    path, object: null}}
    water: {require: {sapling: 1}, receive: {drink: 1}, leaves: {
    material: lava, object: {skeleton: 0.1}}}
```

Here is the Crafter setting:

```
collect:
    tree: {require: {}, receive: {wood: 1}, leaves: {material:
    grass, object: null}}
    stone: {require: {wood_pickaxe: 1}, receive: {stone: 1},
    leaves: {material: path, object: null}}
    coal: {require: {wood_pickaxe: 1}, receive: {coal: 1}, leaves:
     {material: path, object: null}}
    iron: {require: {stone_pickaxe: 1}, receive: {iron: 1}, leaves
    : {material: path, object: null}}
    diamond: {require: {iron_pickaxe: 1}, receive: {diamond: 1},
    leaves: {material: path, object: null}}
    water: {require: {}, receive: {drink: 1}, leaves: {material:
    water, object: null}}
    lava: {require: {}, receive: {drink: 1}, leaves: {material:
    lava, object: null}}
    grass: {require: {}, receive: {sapling: {amount: 1,
    probability: 0.1}}, leaves: {material: grass, object: null}}
    sand: {require: {}, receive: {}, leaves: {material: sand ,
    object: null}}
```

**Placing** For placing achievements, we focus on the ignitability of materials while keeping the requirements for placing stone and saplings unchanged, as these tasks do not involve crafting. To this end, we add the attribute of ignitability for wood, stone, coal, iron, and diamond. We randomly sample the value from [True, False] and ensure a mix of flammable and non-flammable materials.

Crafting tables can be made from any material, while furnaces, which are used for smelting, must be crafted from non-flammable substances. For example, if stone is flammable, it cannot be used to make a furnace. Therefore, the materials for crafting tables can be sampled from wood, stone, coal, iron, and diamond, while the materials for making furnaces must be selected from non-flammable substances. Additionally, saplings can grow on stone as well as grass (reflecting the possibility that saplings on this planet have exceptionally strong vitality).

Here is the Crafter setting for placing achievements:

```
ignitability:
    wood: true
    coal: true
    iron: true
    diamond: false
    stone: false
place:
    stone: {uses: {stone: 1}, where: [grass, sand, path, water,
    lava], type: material}
    table: {uses: {wood: 2}, where: [grass, sand, path], type:
    material}
    furnace: {uses: {stone: 4}, where: [grass, sand, path], type:
    material}
    plant: {uses: {sapling: 1}, where: [grass], type: object}
```

Here is one example of modified placing tasks:

```
ignitability:
    wood: true
    coal: true
    iron: false
    diamond: true
    stone: false
place:
    stone: {uses: {stone: 1}, where: [grass, sand, path, water,
    lava], type: material}
    table: {uses: {wood: 2}, where: [grass, sand, path], type:
    material}
    furnace: {uses: {iron: 4}, where: [grass, sand, path], type:
    material}
    plant: {uses: {sapling: 1}, where: [grass, sand, path, water,
    lava, stone, coal, iron, diamond], type: object}
```

**Crafting**  Regarding crafting achievements, we assume that the names of items often reflect their materials. Thus, we do not alter the raw materials used for tools. Based on the ignitability of the material, we only consider whether a table or furnace is required. For items that are ignitable, both a table and a furnace are required, whereas for non-flammable items, a table suffices.

Here is the Crafter setting for placing achievements:

```
make:
    wood_pickaxe: {uses: {wood: 1}, nearby: [table], gives: 1}
    stone_pickaxe: {uses: {wood: 1, stone: 1}, nearby: [table],
    gives: 1}
    iron_pickaxe: {uses: {wood: 1, coal: 1, iron: 1}, nearby: [
    table, furnace], gives: 1}
    wood_sword: {uses: {wood: 1}, nearby: [table], gives: 1}
    stone_sword: {uses: {wood: 1, stone: 1}, nearby: [table],
    gives: 1}
    iron_sword: {uses: {wood: 1, coal: 1, iron: 1}, nearby: [table
    , furnace], gives: 1}
```

Here is one example of modified crafting tasks:

```
ignitability:
    wood: true
    coal: true
    iron: false
    diamond: true
    stone: false
make:
    wood_pickaxe: {uses: {wood: 1}, nearby: [table, furnace],
    gives: 1}
    stone_pickaxe: {uses: {wood: 1, stone: 1}, nearby: [table,
    furnace], gives: 1}
    iron_pickaxe: {uses: {wood: 1, coal: 1, iron: 1}, nearby: [
    table, furnace], gives: 1}
    wood_sword: {uses: {wood: 1}, nearby: [table, furnace], gives:
     1}
    stone_sword: {uses: {wood: 1, stone: 1}, nearby: [table,
    furnace], gives: 1}
    iron_sword: {uses: {wood: 1, coal: 1, iron: 1}, nearby: [table
    , furnace], gives: 1}
```

### A.6 Key considerations for modification

In addition to several strict principles to prevent the new world from collapsing, we also implemented other measures. Specifically:

- We do not roughly make sweeping changes to the entire Crafter world. Instead, we selectively modify specific types and numbers of elements to control the difficulty. Generally, modifying only the terrain (e.g., water nearby stone instead of sand) is the simplest. Modifying survival settings (e.g., zombies can shoot) presents a moderate challenge, while altering task dependencies (e.g., mining stone yields diamonds) is the most difficult. The more rules we modify, the greater the challenge.

- Additionally, we meticulously consider the content of these rule modifications. While they may seem counter-intuitive, most of them remain reasonable and plausible. For instance, having stone near water is possible, as in cave systems where water of underground lakes or streams often flows over stone. Similarly, zombies infected by a virus might shoot; consuming overly salty beef could increase thirst; a frenzied cow might attack humans; and trees could grow rapidly, with a new tree sprouting immediately after the original one is cut down.

- Furthermore, we specifically invite skilled Crafter players to test the seven chosen experimental worlds. After five episodes of learning and adaptation, these players achieved rewards in the range of 16-18 out of 22 possible achievements. This demonstrates that while our benchmark is challenging, it is also reasonable.

## B  Evaluating Crafter's knowledge of GPT-4

LLMs are pre-trained on vast and diverse textual data, which provides them with extensive world knowledge and commonsense information. This knowledge often aligns with the mechanisms of the Crafter [Hafner, 2021] game, which is why many studies leverage the commonsense knowledge encoded in LLMs to guide RL for more efficient exploration in Crafter. For instance, ELLM [Du et al., 2023] shapes rewards towards commonsense and useful behaviors through a pretrained LLM, while AdaRefiner [Zhang and Lu, 2024] uses sub-goals suggested by the LLM to guide exploration.

To further validate the LLMs' understanding of Crafter's game mechanics, we conduct two additional experiments:

**Knowledge Mastery Quiz** : To assess whether LLMs have internalized Crafter's knowledge, we design a quiz consisting of 72 multiple-choice questions on Crafter's world knowledge. GPT-4 achieves an 81% accuracy rate, indicating that LLMs encode a significant portion of Crafter's game knowledge.

```
Prompt:
I will give you a multiple-choice question to test your
commonsense knowledge. Please choose the correct answer from the
options. If you do not know the answer, please output "I don't
know". The response format is below:
Reasoning: {your reasoning}
Answer: {your answer}
```

**In-Context Knowledge Experiment** : We also experiment with the ReAct model by providing it with Crafter's game knowledge in-context. Results are presented in Table 4. We observe minimal performance improvement with this knowledge compared to without it. Performance dropped further when transitioning from the Default to Mars scenarios, highlighting the challenges of adapting to novel situations. Interestingly, when modifying "Task. Dep" type, providing knowledge led to poorer performance, which may be due to the emphasis on in-context commonsense knowledge making it more difficult to process counter-commonsense situations, further disrupting its ability to perform situated inductive reasoning.

Table 4: Rewards across different worlds with and without Crafter's knowledge.

|  | Default | Terrain | Survival | Task. Dep | Terr. Surv. | Terr. Task. | Surv. Task | All three. |
|---|---|---|---|---|---|---|---|---|
| with knowledge | $7.9^{\pm2.7}$ | $7.8^{\pm3.1}$ | $7.0^{\pm4.1}$ | $1.8^{\pm0.5}$ | $6.8 \pm 1.7$ | $4.4^{\pm0.9}$ | $0.8^{\pm0.5}$ | $0.1^{\pm0.8}$ |
| w/o. knowledge | $7.7^{\pm1.6}$ | $7.4^{\pm2.7}$ | $6.4^{\pm3.7}$ | $5.0^{\pm2.1}$ | $6.7^{\pm2.5}$ | $4.8^{\pm2.0}$ | $1.5^{\pm1.3}$ | $0.7^{\pm1.6}$ |

## C   Pipeline of Skill Library

In this section, we introduce the revised pipeline of **Skill Library**. Based on JARVIS-1 and Voyager [Wang et al., 2023a,b], we further simplify the framework to adapt to our environment. Specifically, given the agents observation (location, inventory, nearby blocks) and task list, we prompt the LLM as a *task proposer* to select a feasible and novel task. Then, the LLM-based *planner* decomposes this high-level task into a sequence of subgoals. The LLM-based *controller* executes these subgoals sequentially by outputting available actions (e.g., move left, place table). However, if the controller outputs "failed" or believes it "succeeded" but the task cannot be accomplished (as indicated by the environments feedback), it suggests that the initial plan provided by the planner may contain errors or that the controller experienced execution failures. Then, the *explainer* tries to identify the errors and re-plan the current task. For successful plans, we store in the skill library along with the task and the agent situation for future reuse in similar situations. Here, task proposer, planner, explainer, and controller are fulfilled by the LLMs.

## D   More results of DreamerV3

We further test Mars using the model trained in Crafter. The results are shown in Table 7. From the results, we observe that DreamerV3 performs well in Default (the same world as training). However, when adapting to a new world that includes partial counter-commonsense elements, the performance drops significantly. These results indicate that the state-of-the-art RL-based method DreamerV3 struggles to quickly adapt to environments with even minor differences (*e.g.*, the "Terrain" world achieves a reward of only 5.3), demonstrating that it does not solve the situated inductive reasoning problem.

Table 5: Results of worlds in Mars using DreamerV3 trained in Crafter.

|  | Default | Terrain | Survival | Task. Dep | Terr. Surv. | Terr. Task. | Surv. Task | All three. |
|---|---|---|---|---|---|---|---|---|
| Reward | $11.5^{\pm1.6}$ | $5.3^{\pm3.4}$ | $6.4^{\pm4.4}$ | $3.0^{\pm2.1}$ | $3.8^{\pm2.6}$ | $3.5^{\pm0.9}$ | $2.2^{\pm2.0}$ | $1.2^{\pm1.3}$ |
| Score (%) | $14.2^{\pm1.3}$ | $6.8^{\pm2.8}$ | $8.7^{\pm4.6}$ | $3.4^{\pm1.3}$ | $3.8^{\pm0.1}$ | $2.4^{\pm0.4}$ | $2.3^{\pm2.1}$ | $1.1^{\pm0.5}$ |

# E    More results of ELLM

In this section, we conduct experiments using ELLM [Du et al., 2023], which leverages LLMs for reward design. To ensure consistency with our setup, we include both intrinsic rewards and health rewards during training. For other hyperparameters, we use the default settings provided in their code[3]. The results are shown in Table 6. The performance of ELLM drops in Mars compared to the Crafter (Default) environment, which aligns with the findings using both RL-based and LLM-based methods. These results suggest that while LLM priors can guide RL exploration, when transferring to a novel world with different game mechanics and knowledge, LLMs struggle due to their lack of situated inductive reasoning. This further validates the difficulty of our Mars benchmark under current methods, underscoring the need for more advanced AI systems that can adapt and reason contextually in novel environments.

Table 6: Results across different worlds with ELLM model.

|  | Default | Terrain | Survival | Task. Dep | Terr. Surv. | Terr. Task. | Surv. Task | All three. |
|---|---|---|---|---|---|---|---|---|
| Reward | $5.0^{\pm 0.5}$ | $4.0^{\pm 1.2}$ | $4.6^{\pm 2.5}$ | $3.3^{\pm 0.8}$ | $2.5^{\pm 0.6}$ | $4.1^{\pm 1.2}$ | $2.4^{\pm 2.2}$ | $1.0^{\pm 0.9}$ |

# F    More results of the open-source model

We conduct additional experiments with the open-source model LLaMA-3.1-8B-instruct. We evaluate both the ReAct and IfR models across different worlds, using the same prompts and hyperparameters as with GPT-4. The results show that LLaMA's performance declines when encountering the Mars environment. Additionally, our model IfR consistently outperforms ReAct across all scenarios. These findings align with the results obtained using GPT-4, further validating the importance of inductive reasoning and highlighting the challenges posed by our benchmark.

Table 7: Results of ReAct and IfR model using LLaMA-3.1-8B-instruct.

|  | Default | Terrain | Survival | Task. Dep | Terr. Surv. | Terr. Task. | Surv. Task | All three. |
|---|---|---|---|---|---|---|---|---|
| ReAct | $3.6^{\pm 2.1}$ | $2.1^{\pm 2.2}$ | $2.3^{\pm 2.5}$ | $2.3^{\pm 1.0}$ | $1.1^{\pm 1.4}$ | $3.0^{\pm 1.6}$ | $0.7^{\pm 2.0}$ | $0.2^{\pm 1.2}$ |
| Ours (IfR) | $3.8^{\pm 2.4}$ | $3.8^{\pm 2.1}$ | $3.7^{\pm 2.8}$ | $2.9^{\pm 1.0}$ | $3.8^{\pm 2.0}$ | $3.3^{\pm 1.2}$ | $1.1^{\pm 1.3}$ | $0.8^{\pm 1.4}$ |

# G    Failure cases of Skill Library

Skill Library's memory only involves successful subgoal sequences, making it difficult to be aware of the real situation for task completion. Consequently, the incorrect path will be reused repetitively. For example, for the task "place table", if the planner outputs "step1: mine tree; step2: place table" and the LLM-based controller accidentally mines grass and obtains wood, the Skill Library will save the incorrect plan for "place table". This reuse of erroneous plans significantly reduces the model's exploration efficiency.

Here are the plans extracted from Skill Library, which is from "Task. Dep" modified world:

---

[3]https://github.com/yuqingd/ellm/

```
skill library:
"collect_wood": [
{ "init_inventory": {},
"table_in_view": false,
"furnace_in_view": false,
"plan": [
"explore("right", 2) # explore to the right to find a tree",
"mine("tree", 1) # mine the tree block to get wood"
]
},
{
"init_inventory": {},
"table_in_view": false,
"furnace_in_view": false,
"plan": [
"mine("tree", 1) # mine the tree block to get wood",
"make("wood sword") # craft a wood sword to attack the cow"
]
}
```

In fact, collecting wood is achieved by mining grass. However, from the example, we can observe the plans extracted from Skill Library is consistent with the default Crafter world. We can infer that the controller unintentionally mined grass and obtained wood, and the Skill Library mechanically saved this plan instead of truly learning the "mining grass yields wood" rule.

## H   Examples of induced rules

**Induced rules:**
1. Interacting with water blocks replenishes the player's drink status.
2. Standing on the iron can increase the player's health.
3. The player can use the table and wood to craft a wood pickaxe.
4. The player can move left on the path.
5. ......

## I   More detailed analysis and case studies

The results in Table 2 have indicated that our benchmark, Mars, is challenging for current methods primarily due to their lack of situated inductive reasoning ability. This ability encompasses two key aspects: inductive reasoning, the ability to summarize observations into abstract "conclusions" that go beyond prior experiences, and situated reasoning, which requires understanding situations dynamically and reasoning with present knowledge accordingly. We provide experimental analyses on both situated reasoning and inductive reasoning separately in Section 3.5. Here, we would like to reiterate and further justify the underlying reasons with sampled cases.

**For inductive reasoning:**     To evaluate it, we measure the precision and recall of the rules predicted by IfR using a GPT-4 evaluator (refer to Figure 5). After five episodes of learning, the precision of rules reached a maximum of only 0.68, with recall not exceeding 0.28. We delve into specific cases to identify two potential reasons for this:

- Firstly, the LLMs' inherent priors limit its exploration space to commonsense domains rather than encouraging open-ended exploration. For example, the model failed to induce the rule "Collecting from diamond without any tools yields 1 coal". According to commonsense scenarios, mining diamonds requires crafting an iron pickaxe. When the inventory lacks an iron pickaxe, the model does not attempt to mine diamonds, thus missing the opportunity to induce this rule.
- Secondly, the model is not truly performing inductive reasoning but is instead relying on retrieving prior knowledge for predictions. Continuing with the previous example, even

when the model accidentally triggers the event of mining a diamond and receiving coal, it induces the incorrect rule "Interacting with a diamond block collects the diamond." Another example is the rule "Interacting with a stone block resulting in stone if the player has a wood pickaxe in their inventory," whereas the actual rule should be "Interacting with a stone block yields diamonds."

**For situated reasoning:**    We evaluate this by providing the game rules of each world in context (refer to Table 3). We observe that LLMs perform better when provided with the necessary rules. However, under counter-commonsense conditions, the improvement is lower than in the default (commonsense) scenarios. We present some failure cases to further analyze this aspect. We find that LLMs struggle to apply world-specific rules to perform thinking and reasoning. Even when they recognize new game mechanisms in the world, they still stubbornly rely on prior knowledge during decision-making instead of really perform situated reasoning in novel scenior. Below are detailed analysis:

> **Case 1:**   The player has successfully collected more coal, which is a valuable resource for crafting and placing items like a furnace.

In this new world, coal should be used to place a table, while LLMs mistakenly assume that coal is used for crafting a furnace. This indicates that the LLMs are not engaging in situated reasoning but are still relying on prior knowledge to retrieve information.

> **Case 2:**   Observation: I took action do (interact with the front water block).
> I am on the grass.
> I see: (object with coordinate)
> water is in front of me.
> <iron(-1, 0), iron(1, 0), water(0, -1), grass(0, 1), grass(-1, -1), grass(1, -1), tree(-1, 1), tree(1, 1), table(2, 0), skeleton(-2, 1)>
> My status: <health: 3/9, food: 9/9, drink: 1/9, energy: 3/9>
> My inventory: <wood: 1, coal: 1, diamond: 3, wood_pickaxe: 1>
> Reflecter: After a second interaction with the water block, the status remains unchanged from the previous attempt, indicating that interacting with the water block directly again did not alter the outcome... it seems that further attempts to interact with the water block in the same manner may not yield different results...

When the agent repeatedly attempts to interact with the water block without any changes occurring, the reflection does not identify the true underlying cause (i.e., drinking water requires a sapling). Instead, the agent simply repeats the facts.

> **Case 3:**   Planner: The task is to make a wood pickaxe. The plan is below:
> # step 1: place("table") # place the crafting table on the ground
> # step 2: make("wood pickaxe") # craft a wood pickaxe using wood from the inventory
> ...
> Reflecter: Since the goal is to place a table, and placing a table consumes 2 diamonds from the player's inventory according to the game mechanics, the first step is to ensure we have the necessary resources to craft a table, which in this case, are diamonds.
> RePlanner: The plan to make a wood pickaxe is below: # step 1: mine("grass") # mine the grass block to potentially get more wood
> # step 2: place("table") # place the crafting table on the ground
> # step 3: make("wood pickaxe") # craft a wood pickaxe using wood from the inventory

Although LLMs can sometimes identify errors through reflection, such as needing to mine diamonds to place the table, when replanning, they still attempt to gather more wood to place the table. This shows that their planning is still influenced by prior knowledge.

> **Case 4:** You are on the grass.
> You see: (object with coordinate)
> grass is in front of you.
> <grass(-1, 0), grass(1, 0), sand(0, -1), grass(0, 1), lava(-1, -1), water(1, -1), grass(-1, 1), sand(1, 1), stone(4, -1), diamond(4, -2)>
> Your status: <health: 6/9, food: 9/9, drink: 8/9, energy: 7/9>
> Your inventory: <sapling: 1, coal: 2, diamond: 1>
> THINK: The player has been moving around the same area without finding new resources or a tree... Given the game mechanics, it's essential to explore new areas to find a tree for wood, which is crucial for crafting tools and advancing in the game. The player's inventory contains a sapling..., but without wood, the options for crafting are limited. The current strategy should involve moving in directions that haven't been explored yet or where the player hasn't been in a while, aiming to uncover different terrains or resources...

The LLM agent continues to search for a tree, when in fact, the nearby stone can be mined without tools. The LLMs need to fully utilize the world-specific rules and the current situation to make optimal decisions, rather than use prior experience to perform inefficient exploration.

## J Compute Resource Details

For running all experiments, we use the hardware resources as listed in Table 8.

Table 8: Compute Resource Details

| CPU | GPT | RAM |
|---|---|---|
| AMD Ryzen 9 5950X@3.4GHz | Nvidia RTX 3090 (24GB) | 64GB |
| AMD EPYC 7642@2.3GHz | Nvidia A100 (40GB) | 1.0T |

## K Licenses

In our code, we have used the following libraries which are covered by the corresponding licenses:

- Crafter (MIT license)
- OpenAI GPT (CC BY-NC-SA 4.0 license)
- Stable Baselines3 (MIT license)
- DreamerV3 (MIT License)

# L Prompt

## L.1 ReAct

**Instruction:** You are playing a new [counter-commonsense] game, where some game mechanics are different from Minecraft. Please unlock as many achievements as possible while ensuring your survival.

Available actions are < move_left, move_right, move_up, move_down, do, sleep, place_stone, place_table, place_furnace, place_plant, make_wood_pickaxe, make_stone_pickaxe, make_iron_pickaxe, make_wood_sword, make_stone_sword, make_iron_sword >, where 'do' means to interact the block at front of the player, including mine the block, attack the creature, and drink.

Unlock the following achievements < Collect Coal, Collect Diamond, Collect Drink, Collect Iron, Collect Sapling, Collect Stone, Collect Wood, kill Skeleton, kill Zombie, kill Cow, Eat Plant, Make Iron Pickaxe, Make Iron Sword, Make Stone Pickaxe, Make Stone Sword, Make Wood Pickaxe, Make Wood Sword, Place Furnace, Place Plant, Place Stone, Place Table, Wake Up >

I will give you in-game observations:
You are on: ...
You see (objects with coordinates): ...
Your status (xx/9):
- health higher than 6 means you're healthy;
- food higher than 6 means you're not hungry;
- drink higher than 6 means you're not thirsty;
- energy higher than 6 means you're not fatigued.
Your inventory (xx/9): ...
You should then respond to me with Thought or Action. You must follow the following criteria:
1) Act as a mentor and guide me on what to do based on my current progress. Do not ask questions or give unmeaningful answers.
2) Ensure your survival, including maintaining health, food, drink, and energy levels.
3) The next task should not be too hard since you may not have the necessary resources or have learned enough skills to complete it yet.
4) When necessary items are not around, explore the map extensively. You should not be doing the same thing over and over again.
5) You may sometimes need to repeat some tasks if you need to collect more resources to complete more difficult tasks. Only repeat tasks if necessary.
6) You should choose available and feasible action.
7) Sleep until the energy is full; you will wake up automatically..
8) When you need to craft tools with table or furnace, if there is table or furnace in the view, please move your position to not more than 2 steps away from it.
9) If both a table and furnace are needed, place them together.

If you respond with Thought, you should only respond in the format: THINK: ...
If you respond with Action, you should only respond in the format: ACTION: ...

## L.2 Reflexion

**Instruction:** You are a good analyst of a new [counter-commonsense] game, where some game mechanics are different from Minecraft.

Available actions are < move_left, move_right, move_up, move_down, do, sleep, place_stone, place_table, place_furnace, place_plant, make_wood_pickaxe, make_stone_pickaxe, make_iron_pickaxe, make_wood_sword, make_stone_sword, make_iron_sword >, where 'do' means to interact the block at front of the player, including mine the block, attack the creature, and drink.

You will be provided with the history of past experiences, including each step's action, reward, score, observations, status information, inventory of the player.

When you reflect, you must follow the following criteria:
1) Determine the tasks the player is trying to accomplish.
2) If the player successfully accomplished the task, extract key learnings and skills; if unsuccessful, provide an explanation of the execution failure according to the current inventory information of the agent and adapt the plan.
3) Analyze changes in rewards and scores: rewards indicate the player's health status and task achievements; scores indicate task diversity. Your goal is to maximize both rewards and scores.
You should only respond in the format: REFLECTION: ...
*{history trajectory}*
reward: *{reward}*
score: *{score}*

## L.3 Skill library

### L.3.1 Task proposer

**Instruction:** You are a helpful assistant trying to play a new [counter-commonsense] 2D game, where some game mechanics are different from Minecraft. Please choose the next task from the task pool to do in the new game. Your ultimate goal is to discover as many diverse things as possible, accomplish as many diverse tasks as possible while ensuring survival, and become the best player in the world.

Task pool: [collect coal, collect diamond, collect drink, collect iron, collect sapling, collect stone, collect wood, kill skeleton, kill zombie, kill cow, eat plant, make iron pickaxe, make iron sword, make stone pickaxe, make stone sword, make wood pickaxe, make wood sword, place furnace, place plant, place stone, place table, wake up]

I will give you the following information:
Player's in-game observation: including the player's status, nearby blocks, and the inventory.
Completed tasks so far: ...
Failed tasks: ... Based on this information, you should propose the next task for the player to do. Follow the criteria below: 1) The task should be diverse and challenging, but not too hard. It should be something that the player can accomplish in the next few steps.
2) You may sometimes need to repeat some tasks if you need to collect more resources to complete more difficult tasks. Only repeat tasks if necessary.
3) The task should be related to the player's current status, nearby blocks, and inventory.

You should only respond in the format described below: RESPONSE FORMAT:
Reasoning: Based on the information I listed above, do reasoning about what the next task should be.
Task: The next task.

Here are some examples: *{examples}*

### L.3.2 Task planner

**Instruction:** You are a helper agent in a new [counter-commonsense] 2D game, where some mechanics are different from Minecraft. Based on your current inventory and observations, you need to generate sequences of subgoals for a certain task. Please refer to the history dialogue to give the plan consisting of templates. Do not explain or give any other instructions.

You must follow the criteria below:
1) You should only mine [stone, coal, iron, tree, diamond, water, lava, grass, sand, ripe-plant] blocks.
2) You should only attack movable creatures.
3) You should only place [stone, table, furnace, sapling] blocks.
4) You should only craft [wood pickaxe, stone pickaxe, iron pickaxe, wood sword, stone sword, iron sword] tools.
5) You should choose available subgoals to complete the task.
6) You are probably provided some past successful plans to refer to.
8) Not all creatures are friendly. When you are attacked, please attack back.
9) You should only perform the subgoals that are feasible based on the current inventory and observations.
10) This is a 2D game, so when you encounter an obstacle, you should mine it or place a block to build a "path" or make a detour.

Here are some subgoals for reference:
mine(block_name, amount) # mine a specified amount of blocks of the block_name.
attack(creature, amount) # attack the specified number of creatures that can move. Creatures include zombies, skeletons, cows, etc.
sleep(); # put the player to sleep.
place(block_name); # place the block. Note that you do not need to craft tables and furnaces; you can place them directly.
make(tool_name); # craft a tool.
explore(direction, steps); # the player explores in the specified direction for the given steps.

Here are some examples: *{examples}*

### L.3.3 Explainer

**Instruction:** You are a helpful assistant trying to play a new [counter-commonsense] 2D game, where some mechanics are different from Minecraft. Here are some actions that the agent fails to perform in the game. Please give an explanation of action execution failure according to the current inventory information of the agent and history dialogue.

You must follow the criteria below: 1) You should only mine [stone, coal, iron, tree, diamond, water, lava, grass, sand, ripe-plant] blocks.
2) You should only attack movable creatures.
3) You should only place [stone, table, furnace, sapling] blocks.
4) You should only craft [wood pickaxe, stone pickaxe, iron pickaxe, wood sword, stone sword, iron sword] tools.
5) Not all creatures are friendly. When you are attacked, please attack back.
6) This is a 2D game, so when you encounter an obstacle, you should mine it or place a block to build a "path" or make a detour.
7) In the new game, it is possible that some tasks or creatures are different from Minecraft. For example, you may need some tools to mine a tree block. Thus, when you attempt to accomplish a task multiple times but fail, please try to explore more counter-commonsense knowledge.

Here are some examples: *{examples}*

### L.3.4 Replanner

> **Instruction:** Please fix the above errors and replan the task [*{task}*]

### L.3.5 Controller

> **Instruction:** You are a helpful assistant trying to play a new [counter-commonsense] 2D game, where some mechanics are different from Minecraft. Given the current observation and the goal, you need to generate the action to complete the goal. You can only perform the following actions:
>
> Available actions are < move_left, move_right, move_up, move_down, do, sleep, place_stone, place_table, place_furnace, place_plant, make_wood_pickaxe, make_stone_pickaxe, make_iron_pickaxe, make_wood_sword, make_stone_sword, make_iron_sword >, where 'do' means to interact with the block in front of the player, including mining the block, attacking creatures, and drinking; "SUCCEED" means that the goal is achieved; "FAILED" means that it is too hard to achieve the goal.
>
> You should follow the criteria below:
> 1) When the desired item is not immediately visible, it is essential to explore the surroundings to locate it. You can move strategically in the direction where the item is likely to be found.
> 2) Not all creatures are friendly. When you are attacked, please attack back.
> 3) When you need to craft tools with a table or furnace, if there is a table or furnace in view, move your position to not more than 2 steps away from it.
> 4) When a table and furnace are needed simultaneously, place them together and place them on proper terrain.
> 5) This is a 2D game, so when you encounter an obstacle, you should mine it or place a block to build a "path" or find a detour.
> 6) When you mine a block, attack a creature, or drink, you must face the block.
> 7) If you move left, your x-coordinate will decrease by 1; if you move right, your x-coordinate will increase by 1; if you move up, your y-coordinate will increase by 1; if you move down, your y-coordinate will decrease by 1.
>
> You should only respond in the format described below:
> RESPONSE FORMAT:
> Reasoning: Based on the information I listed above and history dialogue, do reasoning about how to achieve the goal.
> Action: The next action.
> Here some examples: *{examples}*
> subgoal: *{subgoal}*

## L.4 Induction from Reflection

> **Instruction:** You are a helpful assistant with inductive reasoning. Given the history trajectory, including actions and observations, you need to reflect on the action execution results and determine the possible mechanism of the new game. The mechanism should be consistent with the game rules and the player's inventory information.
>
> You should only respond in the format described below:
> RESPONSE FORMAT:
> Reasoning: Based on the information I listed above and history dialogue, do reasoning about the mechanism of the new game.
> Mechanism: The mechanism of the new game.
>
> Here are some examples: *{examples}*
> *{history trajectory}*

## L.5 Few-shot demonstrations in IfR module

In fact, we use few-shot induction examples to prompt LLM for inductive reasoning. Here are the few-shot examples we provided:

> Reasoning: The player's health decreased by 2 after shot by arrow, indicating that the arrow of skeleton is harmful to the player.
> Mechanism: The arrow of skeleton can cause damage to the player.
>
> Reasoning: The player is facing the water block and cannot enter the water block, indicating that the player cannot swim or the water block is not walkable.
> Mechanism: the water block is not walkable.
>
> Reasoning: The player has been mining the stone block for a long time but has not yet obtained the stone, indicating that the stone block cannot be mined by hand.
> Mechanism: The stone block cannot be mined by hand.
>
> Reasoning: The player has been placing the table in the front stone block for a long time but has not yet placed the table, indicating that the table cannot be placed on the stone block.
> Mechanism: The table cannot be placed on the stone block.
>
> Reasoning: The player can place the table in the front grass block, indicating that the table can be placed on the grass block.
> Mechanism: The table can be placed on the grass block.

Despite providing these few-shot induction demonstrations, LLMs still perform poorly in inducing new rules in novel scenarios. This is likely because in-context learning is heavily dependent on the similarity of provided examples to the target task and the coherence of data distribution [Work, Liu et al., 2021, Zhao et al., 2021, Akyürek et al., 2022]. When LLMs are required to induce rules that are distinct from the examples in a novel scenario, it becomes difficult for them to perform inductive reasoning effectively through in-context learning.

# M Configurations of seven worlds in Mars

## M.1 Terrain

The world "Terrain" only changes the terrain distribution element.

```
terrain_neighbour:
    coal: grass
    iron: sand
    diamond: stone
    lava: stone
    tree: path
    player: sand
    water: stone
```

## M.2 Survival

The world "Survival" only changes the survival setting.

```
npc_objects:
  cow:
    eatable: false
    defeatable: true
    arrowable: true
    closable: false
```

```
    can_walk: true
    closable_health_damage_func: 0
    attackable: true
    eat_health_damage_func: 0
    inc_food_func: 0
    inc_thirst_func: 0
    arrow_damage_func: -1
  zombie:
    eatable: true
    defeatable: false
    arrowable: false
    closable: true
    can_walk: true
    closable_health_damage_func: 0
    attackable: true
    eat_health_damage_func: 1
    inc_food_func: 1
    inc_thirst_func: 1
    arrow_damage_func: 0
  skeleton:
    eatable: true
    defeatable: false
    arrowable: false
    closable: false
    can_walk: true
    closable_health_damage_func: 0
    attackable: false
    eat_health_damage_func: -1
    inc_food_func: -1
    inc_thirst_func: -1
    arrow_damage_func: 0
  plant:
    eatable: true
    defeatable: false
    arrowable: false
    closable: false
    can_walk: true
    closable_health_damage_func: 0
    attackable: false
    eat_health_damage_func: 0
    inc_food_func: 1
    inc_thirst_func: 1
    arrow_damage_func: 0
drink:
  water:
    inc_drink_func: 1
    inc_damage_func: -1
    inc_food_func: 0
  lava:
    inc_drink_func: -1
    inc_damage_func: -1
    inc_food_func: 1
```

## M.3   Task. Dep

The world "Task. Dep" only changes the task dependency element.

```
ignitability:
  wood: true
```

```
  coal: true
  iron: false
  diamond: true
  stone: false
collect:
  tree: {require: {iron_pickaxe: 1}, receive: {stone: 1}, leaves:
  {material: grass, object: null}}
  stone: {require: {}, receive: {diamond: 1}, leaves: {material:
  grass, object: null}}
  coal: {require: {wood_pickaxe: 1}, receive: {iron: 1}, leaves: {
  material: path, object: null}}
  iron: {require: {stone_pickaxe: 1}, receive: {sapling: {amount:
  1, probability: 0.1}}, leaves: {material: path, object: null}}
  diamond: {require: {}, receive: {coal: 1}, leaves: {material:
  path, object: null}}
  water: {require: {}, receive: {drink: 1}, leaves: {material:
  water, object: {zombie: 0.1}}}
  lava: {require: {}, receive: {drink: 1}, leaves: {material: lava
  , object: null}}
  grass: {require: {}, receive: {wood: 1}, leaves: {material:
  grass, object: null}}
  sand: {require: {}, receive: {}, leaves: {material: sand, object
  : null}}
place:
  stone: {uses: {stone: 1}, where: [grass, sand, path, water, lava
  ], type: material}
  table: {uses: {diamond: 2}, where: [grass, sand, path], type:
  material}
  furnace: {uses: {iron: 4}, where: [grass, sand, path], type:
  material}
  plant: {uses: {sapling: 1}, where: [grass, sand, path, water,
  lava, stone, coal, iron, diamond], type: object}
make:
  wood_pickaxe: {uses: {wood: 1}, nearby: [table], gives: 1}
  stone_pickaxe: {uses: {wood: 1, stone: 1}, nearby: [table,
  furnace], gives: 1}
  iron_pickaxe: {uses: {wood: 1, coal: 1, iron: 1}, nearby: [table
  ], gives: 1}
  wood_sword: {uses: {wood: 1}, nearby: [table], gives: 1}
  stone_sword: {uses: {wood: 1, stone: 1}, nearby: [table, furnace
  ], gives: 1}
  iron_sword: {uses: {wood: 1, coal: 1, iron: 1}, nearby: [table],
   gives: 1}
```

### M.4 Terr. Surv.

The world "Terr. Surv." involves changeing the terrain and survival setting.

```
terrain_neighbour:
  coal: water
  iron: sand
  diamond: stone
  lava: grass
  tree: path
  player: path
  water: sand
walkable_effect:
  stone: {walkable: true, walk_health: 0, dieable: false}
  diamond: {walkable: false, walk_health: 0, dieable: false}
```

```
  coal: {walkable: true, walk_health: 0, dieable: true}
  iron: {walkable: false, walk_health: 0, dieable: false}
  water: {walkable: true, walk_health: 1, dieable: false}
  lava: {walkable: false, walk_health: 0, dieable: false}
  grass: {walkable: false, walk_health: 0, dieable: false}
  path: {walkable: true, walk_health: 0, dieable: false}
  sand: {walkable: true, walk_health: 1, dieable: false}
  tree: {walkable: false, walk_health: 0, dieable: false}
npc_objects:
  cow:
    eatable: true
    defeatable: false
    attackable: true
    arrowable: false
    closable: false
    can_walk: true
    closable_health_damage_func: -1
    eat_health_damage_func: 0
    arrow_damage_func: 0
    inc_food_func: 0
    inc_thirst_func: 1
  zombie:
    eatable: true
    defeatable: false
    attackable: true
    arrowable: false
    closable: false
    can_walk: true
    closable_health_damage_func: 1
    eat_health_damage_func: 0
    arrow_damage_func: 0
    inc_food_func: 1
    inc_thirst_func: 0
  skeleton:
    eatable: true
    defeatable: false
    attackable: true
    arrowable: true
    closable: true
    can_walk: true
    closable_health_damage_func: -1
    eat_health_damage_func: -1
    arrow_damage_func: 1
    inc_food_func: 0
    inc_thirst_func: 0
  plant:
    eatable: false
    defeatable: true
    attackable: false
    arrowable: true
    closable: false
    can_walk: false
    closable_health_damage_func: -1
    eat_health_damage_func: 0
    arrow_damage_func: 0
    inc_food_func: 0
    inc_thirst_func: 0
drink:
  lava:
```

```
    inc_drink_func: 1
    inc_damage_func: 1
    inc_food_func: -1
  water:
    inc_drink_func: -1
    inc_damage_func: -1
    inc_food_func: 1
```

## M.5   Terr. Task.

The world "Terr. Task." involves changeing the terrain and task dependency.

```
terrain_neighbour:
  coal: path
  iron: path
  diamond: grass
  lava: path
  tree: stone
  player: path
  water: sand
walkable_effect:
  stone: {walkable: true, walk_health: 0, dieable: false}
  diamond: {walkable: false, walk_health: 0, dieable: false}
  coal: {walkable: false, walk_health: 0, dieable: false}
  iron: {walkable: true, walk_health: 1, dieable: false}
  water: {walkable: true, walk_health: -1, dieable: false}
  lava: {walkable: false, walk_health: 0, dieable: false}
  grass: {walkable: true, walk_health: 1, dieable: false}
  path: {walkable: true, walk_health: 0, dieable: false}
  sand: {walkable: true, walk_health: 0, dieable: false}
  tree: {walkable: false, walk_health: 0, dieable: false}
ignitability:
  wood: false
  coal: false
  iron: true
  diamond: false
  stone: true
collect:
  tree: {require: {}, receive: {coal: 1}, leaves: {material: path,
   object: null}}
  stone: {require: {}, receive: {stone: {amount: 1, probability:
  0.5}, wood: {amount: 1, probability: 0.5}}, leaves: {material:
  diamond, object: null}}
  coal: {require: {wood_pickaxe: 1}, receive: {coal: 1}, leaves: {
  material: lava, object: null}}
  iron: {require: {stone_pickaxe: 1}, receive: {iron: 1}, leaves:
  {material: lava, object: null}}
  diamond: {require: {stone_pickaxe: 1}, receive: {diamond: 1},
  leaves: {material: water, object: null}}
  water: {require: {}, receive: {drink: 1}, leaves: {material:
  water, object: {skeleton: 0.1}}}
  lava: {require: {sapling: 1}, receive: {drink: 1}, leaves: {
  material: stone, object: {}}}
  grass: {require: {wood_pickaxe: 1}, receive: {sapling: {amount:
  1, probability: 0.1}}, leaves: {material: grass, object: null}}
  sand: {require: {iron_pickaxe: 1}, receive: {coal: 1}, leaves: {
  material: lava, object: None}}
place:
```

```
  stone: {uses: {stone: 1}, where: [grass, sand, path, water, lava
  ], type: material}
  table: {uses: {stone: 4}, where: [grass, sand, path], type:
  material}
  furnace: {uses: {coal: 4}, where: [grass, sand, path], type:
  material}
  plant: {uses: {sapling: 1}, where: [grass, sand, path, water,
  lava, stone, coal, iron, diamond], type: object}
make:
  wood_pickaxe: {uses: {wood: 1}, nearby: [table], gives: 1}
  stone_pickaxe: {uses: {wood: 1, stone: 1}, nearby: [table,
  furnace], gives: 1}
  iron_pickaxe: {uses: {wood: 1, coal: 1, iron: 1}, nearby: [table
  ], gives: 1}
  wood_sword: {uses: {wood: 1}, nearby: [table], gives: 1}
  stone_sword: {uses: {wood: 1, stone: 1}, nearby: [table, furnace
  ], gives: 1}
  iron_sword: {uses: {wood: 1, coal: 1, iron: 1}, nearby: [table],
   gives: 1}
```

## M.6 Surv. Task

The world "Surv. Task." involves changeing the survival setting and task dependency.

```
npc_objects:
  cow:
    eatable: true
    defeatable: false
    arrowable: false
    closable: true
    can_walk: true
    closable_health_damage_func: -1
    attackable: true
    eat_health_damage_func: 1
    inc_food_func: 1
    inc_thirst_func: 1
    arrow_damage_func: 0
  zombie:
    eatable: false
    defeatable: true
    arrowable: false
    closable: false
    can_walk: true
    closable_health_damage_func: -1
    attackable: true
    eat_health_damage_func: 0
    inc_food_func: 0
    inc_thirst_func: 0
    arrow_damage_func: 0
  skeleton:
    eatable: false
    defeatable: true
    arrowable: false
    closable: true
    can_walk: true
    closable_health_damage_func: 0
    attackable: false
    eat_health_damage_func: 0
    inc_food_func: 0
```

```
    inc_thirst_func: 0
    arrow_damage_func: 0
  plant:
    eatable: true
    defeatable: false
    arrowable: true
    closable: false
    can_walk: true
    closable_health_damage_func: 0
    attackable: false
    eat_health_damage_func: 1
    inc_food_func: 1
    inc_thirst_func: -1
    arrow_damage_func: 1
drink:
  lava:
    inc_drink_func: 1
    inc_damage_func: -1
    inc_food_func: 1
  water:
    inc_drink_func: -1
    inc_damage_func: 1
    inc_food_func: 1
ignitability:
  wood: false
  coal: true
  iron: true
  diamond: true
  stone: false
collect:
  tree: {require: {}, receive: {wood: {amount: 1, probability:
  0.5}, diamond: {amount: 1, probability: 0.5}}, leaves: {material
  : coal, object: null}}
  stone: {require: {}, receive: {stone: 1}, leaves: {material:
  path, object: null}}
  coal: {require: {}, receive: {coal: 1}, leaves: {material: water
  , object: null}}
  iron: {require: {stone_pickaxe: 1}, receive: {iron: 1}, leaves:
  {material: water, object: null}}
  diamond: {require: {iron_pickaxe: 1}, receive: {diamond: 1},
  leaves: {material: diamond, object: null}}
  water: {require: {sapling: 1}, receive: {drink: 1}, leaves: {
  material: lava, object: {skeleton: 0.1}}}
  lava: {require: {sapling: 1}, receive: {drink: 1}, leaves: {
  material: water, object: {zombie: 0.1}}}
  grass: {require: {wood_pickaxe: 1}, receive: {sapling: {amount:
  1, probability: 0.1}}, leaves: {material: iron, object: null}}
  sand: {require: {}, receive: {sapling: 1}, leaves: {material:
  coal, object: {skeleton: 0.1}}}
place:
  stone: {uses: {stone: 1}, where: [grass, sand, path, water, lava
  ], type: material}
  table: {uses: {coal: 4}, where: [grass, sand, path], type:
  material}
  furnace: {uses: {stone: 4}, where: [grass, sand, path], type:
  material}
  plant: {uses: {sapling: 1}, where: [grass, sand, path, water,
  lava, stone, coal, iron, diamond], type: object}
make:
```

```
wood_pickaxe: {uses: {wood: 1}, nearby: [table], gives: 1}
stone_pickaxe: {uses: {wood: 1, stone: 1}, nearby: [table],
gives: 1}
iron_pickaxe: {uses: {wood: 1, coal: 1, iron: 1}, nearby: [table
, furnace], gives: 1}
wood_sword: {uses: {wood: 1}, nearby: [table], gives: 1}
stone_sword: {uses: {wood: 1, stone: 1}, nearby: [table], gives:
 1}
iron_sword: {uses: {wood: 1, coal: 1, iron: 1}, nearby: [table,
furnace], gives: 1}
```

## M.7 All. three (changed)

The world "All. three (changed)" involves changeing terrain, survival setting and task dependency.

```
terrain_neighbour:
  coal: stone
  iron: path
  diamond: sand
  lava: grass
  tree: grass
  player: diamond
  water: iron
walkable_effect:
  stone: {walkable: true, walk_health: 0, dieable: false}
  diamond: {walkable: true, walk_health: 0, dieable: false}
  coal: {walkable: false, walk_health: 0, dieable: false}
  iron: {walkable: true, walk_health: 0, dieable: false}
  water: {walkable: true, walk_health: 0, dieable: true}
  lava: {walkable: false, walk_health: 0, dieable: false}
  grass: {walkable: true, walk_health: 0, dieable: false}
  path: {walkable: false, walk_health: 0, dieable: false}
  sand: {walkable: true, walk_health: -1, dieable: false}
  tree: {walkable: false, walk_health: 0, dieable: false}
npc_objects:
  cow:
    eatable: false
    defeatable: true
    attackable: false
    arrowable: true
    closable: false
    can_walk: false
    closable_health_damage_func: 0
    eat_health_damage_func: 0
    arrow_damage_func: -1
    inc_food_func: 0
    inc_thirst_func: 0
  zombie:
    eatable: true
    defeatable: false
    attackable: true
    arrowable: false
    closable: false
    can_walk: false
    closable_health_damage_func: 1
    eat_health_damage_func: 0
    arrow_damage_func: 0
    inc_food_func: 1
    inc_thirst_func: -1
```

```
  skeleton:
    eatable: false
    defeatable: true
    attackable: false
    arrowable: false
    closable: false
    can_walk: false
    closable_health_damage_func: 0
    eat_health_damage_func: 0
    arrow_damage_func: 0
    inc_food_func: 0
    inc_thirst_func: 0
  plant:
    eatable: true
    defeatable: false
    attackable: true
    arrowable: false
    closable: false
    can_walk: false
    closable_health_damage_func: -1
    eat_health_damage_func: 1
    arrow_damage_func: 0
    inc_food_func: -1
    inc_thirst_func: 1
drink:
  lava:
    inc_drink_func: 1
    inc_damage_func: 0
    inc_food_func: 1
  water:
    inc_drink_func: 1
    inc_damage_func: 0
    inc_food_func: -1
ignitability:
  wood: true
  coal: false
  iron: false
  diamond: false
  stone: true
collect:
  tree: {require: {iron_pickaxe: 1}, receive: {iron: 1}, leaves: {
  material: path, object: null}}
  stone: {require: {}, receive: {wood: {amount: 1, probability:
  0.5}, stone: {amount: 1, probability: 0.5}}, leaves: {material:
  sand, object: null}}
  coal: {require: {wood_pickaxe: 1}, receive: {coal: 1}, leaves: {
  material: stone, object: null}}
  iron: {require: {}, receive: {iron: 1}, leaves: {material: tree,
   object: null}}
  diamond: {require: {stone_pickaxe: 1}, receive: {diamond: 1},
  leaves: {material: stone, object: null}}
  water: {require: {sapling: 1}, receive: {drink: 1}, leaves: {
  material: tree, object: {}}}
  lava: {require: {}, receive: {drink: 1}, leaves: {material: lava
  , object: {skeleton: 0.1}}}
  grass: {require: {wood_pickaxe: 1}, receive: {sapling: {amount:
  1, probability: 0.1}}, leaves: {material: stone, object: null}}
  sand: {require: {wood_pickaxe: 1}, receive: {sapling: 1}, leaves
  : {material: lava, object: {cow: 0.1}}}
```

```
place:
  stone: {uses: {stone: 1}, where: [grass, sand, path, water, lava
  ], type: material}
  table: {uses: {wood: 2}, where: [grass, sand, path], type:
  material}
  furnace: {uses: {iron: 4}, where: [grass, sand, path], type:
  material}
  plant: {uses: {sapling: 1}, where: [grass, sand, path, water,
  lava, stone, coal, iron, diamond], type: object}
make:
  wood_pickaxe: {uses: {wood: 1}, nearby: [table, furnace], gives:
   1}
  stone_pickaxe: {uses: {wood: 1, stone: 1}, nearby: [table],
  gives: 1}
  iron_pickaxe: {uses: {wood: 1, coal: 1, iron: 1}, nearby: [table
  ], gives: 1}
  wood_sword: {uses: {wood: 1}, nearby: [table, furnace], gives:
  1}
  stone_sword: {uses: {wood: 1, stone: 1}, nearby: [table], gives:
   1}
  iron_sword: {uses: {wood: 1, coal: 1, iron: 1}, nearby: [table],
   gives: 1}
```

