# OpenReview forum: "Mars: Situated Inductive Reasoning in an Open-World Environment"
_NeurIPS.cc/2024/Datasets_and_Benchmarks_Track — NeurIPS 2024 Track Datasets and Benchmarks Poster_

### Official Review · Reviewer_V8Vv · 2024-06-23
**Paper 1225 Official Review**

**Rating:** 5
**Confidence:** 3

**Review:**

Please check all the sections below for detailed reviews.

**Strengths:**

- The authors introduce Mars, an interactive environment designed for studying situated inductive reasoning. By modifying terrain, survival settings, and task dependencies, Mars generates diverse worlds. This new dataset provides a valuable testbed for evaluating machine intelligence models' capabilities in acquiring and applying new knowledge in specific contexts.

- The paper addresses the challenging problem of situated inductive reasoning, crucial for developing adaptive AI systems. By emphasizing the situatedness and abstractiveness aspects of inductive reasoning, the authors highlight the need to understand and reason dynamically in different contexts. This problem bridges the gap between pre-stored knowledge and the ability to induce new general knowledge, paving the way for advancements in machine intelligence.

- Experiments on the Mars dataset reveal that existing approaches struggle with situated inductive reasoning. The exploration of Induction from Reflection, where agents reason from historical trajectories, demonstrates superior performance. These findings highlight the importance of inductive reasoning in the Mars environment and provide valuable insights into the limitations of current methods.

**Additional Feedback:**

None

**Clarity:**

The paper is generally well-written and easy to read, with clear explanations. However, there are some grammatical typos and capitalization errors that require further proofreading.

**Correctness:**

The paper presents a benchmark, and the experiments are generally sound and logical.

**Documentation:**

I could not find any dataset documentation in either the paper submission or the supplementary materials. However, both the code and dataset have been released.

**Ethics:**

None that I can think of.

**Limitations:**

The authors explicitly discussed the limitations of their work and proposed potential solutions to address them.

**Opportunities For Improvement:**

While I appreciate the authors' motivation in creating a new world with novel rules and counter-commonsense modifications to challenge the agent's learning capabilities and survival skills, I have a few concerns regarding this setting.

- Firstly, these modifications actually make the world counter-commonsense or counter-intuitive. Despite the authors' implementation of several general rules to prevent the new world from collapsing, there are still potential instances where chaos can arise. This poses a significant challenge for LLMs to behave correctly, as they encounter contradictions between the new world and their internal knowledge, which they cannot rectify themselves. Consequently, the observed results are expected, and I understand the rationale behind the authors' approach. However, I would suggest that the authors curate the benchmark in a way similar to the example described in the abstract. They could select real-world rules with contradictions and present a specific rule to LLMs, allowing them to function as agents in the new environment. This would provide more practical insights for designing inductive agents. Given its current definition, it is evident that the newly curated environment explicitly challenges the agent by presenting counterfactual rules. Although these agents typically struggle to perform well, if instructed with these rules, they show improved performance.

- Furthermore, while the experimental results demonstrate the challenges of the benchmark, it is important to determine the exact sources of these challenges. Are they primarily due to the lack of inductive reasoning ability, difficulties in interacting with the new world, or the complexity of handling uncommonsense scenarios, such as cows shooting? It would be beneficial to have further discussions on these aspects. In addition to the existing analysis, I suggest conducting manual checks on sampled cases to justify the underlying reasons.

- Moreover, I noticed that some implementation details and data statistics are missing. Detailed hyperparameter settings of the baseline methods should be provided, along with reporting the statistics of the newly curated dataset. For instance, when converting visual signals into text for LLM processing, it would be helpful to know the range of token lengths and whether this could potentially overwhelm the LLMs' context.

**Relation To Prior Work:**

The authors have discussed their relation to prior works in a dedicated section, and the comparisons made are comprehensive.

**Summary And Contributions:**

The paper explores the challenge of situated inductive reasoning for machine intelligence. The authors introduce Mars, an interactive environment specifically designed to assess models' capabilities in deriving new general knowledge and applying it to context-sensitive decision-making.

Mars incorporates counter-commonsense game mechanisms and requires agents to actively interact with their surroundings, derive useful rules, and perform tasks in specific contexts.

Experiments conducted on various RL-based and LLM-based methods reveal the difficulty of the situated inductive reasoning benchmark.

The paper also explores Induction from Reflection, where agents perform inductive reasoning from historical trajectories, showing superior performance. The main contributions of this work lie in advancing the field of situated inductive reasoning, providing a testbed for evaluating models' reasoning abilities, and fostering the development of adaptive and context-sensitive AI systems.

---

> ### Author Rebuttal · Authors · 2024-08-16
>
> Thank you for your thorough review and insightful comments on our paper. We appreciate your recognition of the motivation of our work in advancing situated inductive reasoning and your suggestions for improvement. Below, we address your concerns regarding the counter-commonsense world modifications, the sources of challenges in our benchmark, and the additional implementation details.
>
> > W1: Firstly, these modifications actually make the world counter-commonsense or counter-intuitive. Despite the authors' implementation of several general rules to prevent the new world from collapsing, there are still potential instances where chaos can arise... Given its current definition, it is evident that the newly curated environment explicitly challenges the agent by presenting counterfactual rules. Although these agents typically struggle to perform well, if instructed with these rules, they show improved performance.
>
> Thank you for your insightful comment. We would like to clarify that while our benchmark is challenging, it is also carefully designed to be both controllable and reasonable. Specifically:
> 1. We do not roughly make sweeping changes to the entire Crafter world. Instead, we **selectively modify specific types and numbers of elements to control the difficulty**. Generally, modifying only the terrain (e.g., water nearby stone instead of sand) is the simplest. Modifying survival settings (e.g., zombies can shoot) presents a moderate challenge, while altering task dependencies (e.g., mining stone yields diamonds) is the most difficult. The more rules we modify, the greater the challenge.
> 2. Additionally, we meticulously **consider the content of these rule modifications**. While they may seem counter-intuitive, most of them remain reasonable and plausible. For instance, having stone near water is possible, as in cave systems where water of underground lakes or streams often flows over stone. Similarly, zombies infected by a virus might shoot; consuming overly salty beef could increase thirst; a frenzied cow might attack humans; and trees could grow rapidly, with a new tree sprouting immediately after the original one is cut down.
> 3. Furthermore, we specifically **invite skilled Crafter players to test the seven chosen experimental worlds**. After five episodes of learning and adaptation, these players achieved rewards in the range of 16-18 out of 22 possible achievements. This demonstrates that while our benchmark is challenging, it is also reasonable.
>
> We also want to emphasize that our paper introduces the problem of situated inductive reasoning, which requires the agent to dynamically understand the present and live observations, quickly derive new general knowledge (rules) through interactions (induction), and effectively apply the newly acquired knowledge in new scenarios (deduction). Regarding the suggestion to provide world-specific rules, we indeed conduct experiments on this aspect. Please refer to Section 3.5 on situated reasoning. We acknowledge the reviewer's point that even when given the specific rules, although there is some performance improvement, the agents still struggle to fully understand the current situation and effectively apply the rules to decision-making tasks.
>
>
> > W2: ...it is important to determine the exact sources of these challenges...It would be beneficial to have further discussions on these aspects. In addition to the existing analysis, I suggest conducting manual checks on sampled cases to justify the underlying reasons.
>
> Thanks for your valuable feedback regarding more detailed analysis and case studies. We believe that the experimental results indicate that our benchmark, Mars, is challenging for current methods primarily due to their lack of situated inductive reasoning ability. These ability encompasses two key aspects: **inductive reasoning**, the ability to summarize observations into abstract "conclusions" that go beyond prior experiences (Line 40), and **situated reasoning**, which requires understanding situations dynamically and reasoning with present knowledge accordingly (Line 35). We provide experimental analyses on both situated reasoning and inductive reasoning separately in Section 3.5. Here, we would like to reiterate and further justify the underlying reasons with sampled cases.
> 1. For inductive reasoning: To evaluate it, we measure the precision and recall of the rules predicted by IfR using a GPT-4 evaluator (refer to Figure 5). After five episodes of learning, the precision of rules reached a maximum of only 0.68, with recall not exceeding 0.28. We delve into specific cases to identify two potential reasons for this:
>     * Firstly, the LLMs' inherent priors limit its exploration space to commonsense domains rather than encouraging open-ended exploration. For example, the model failed to induce the rule "Collecting from diamond without any tools yields 1 coal". According to commonsense scenarios, mining diamonds requires crafting an iron pickaxe. When the inventory lacks an iron pickaxe, the model does not attempt to mine diamonds, thus missing the opportunity to induce this rule.
>     * Secondly, the model is not truly performing inductive reasoning but is instead relying on retrieving prior knowledge for predictions. Continuing with the previous example, even when the model accidentally triggers the event of mining a diamond and receiving coal, it induces the incorrect rule "Interacting with a diamond block collects the diamond." Another example is the rule "Interacting with a stone block resulting in stone if the player has a wood pickaxe in their inventory," whereas the actual rule should be "Interacting with a stone block yields diamonds."

---

> > ### Author Rebuttal · Authors · 2024-08-16
> >
> > 2. For situated reasoning: We evaluate this by providing the game rules of each world in context (refer to Table 3). We observe that LLMs perform better when provided with the necessary rules. However, under counter-commonsense conditions, the improvement is lower than in the default (commonsense) scenarios. We present some failure cases to further analyze this aspect. We find that LLMs struggle to apply world-specific rules to perform thinking and reasoning. Even when they recognize new game mechanisms in the world, they still stubbornly rely on prior knowledge during decision-making instead of really perform situated reasoning in novel scenior. Below are detailed analysis:
> >     * **Case 1:** In this new world, coal should be used to place a table, while LLMs mistakenly assume that coal is used for crafting a furnace. This indicates that the LLMs are not engaging in situated reasoning but are still relying on prior knowledge to retrieve information.
> >         ```
> >         The player has successfully collected more coal, which is a valuable resource for crafting and placing items like a furnace.
> >         ```
> >
> >     * **Case 2:** When the agent repeatedly attempts to interact with the water block without any changes occurring, the reflection does not identify the true underlying cause (i.e., drinking water requires a sapling). Instead, the agent simply repeats the facts.
> >         ```
> >         Observation: I took action do (interact with the front water block).
> >         I am on the grass.
> >         I see: (object with coordinate)
> >         water is in front of me.
> >         <iron(-1, 0), iron(1, 0), water(0, -1), grass(0, 1), grass(-1, -1), grass(1, -1), tree(-1, 1), tree(1, 1), table(2, 0), skeleton(-2, 1)>
> >         My status: <health: 3/9, food: 9/9, drink: 1/9, energy: 3/9>
> >         My inventory: <wood: 1, coal: 1, diamond: 3, wood_pickaxe: 1>
> >         Reflecter: After a second interaction with the water block, the status remains unchanged from the previous attempt, indicating that interacting with the water block directly again did not alter the outcome... it seems that further attempts to interact with the water block in the same manner may not yield different results... `
> >         ```
> >     * **Case 3:** Although LLMs can sometimes identify errors through reflection, such as needing to mine diamonds to place the table, when replanning, they still attempt to gather more wood to place the table. This shows that their planning is still influenced by prior knowledge.
> >         ```
> >         Planner: The task is to make a wood pickaxe. The plan is below:
> >         # step 1: place("table") # place the crafting table on the ground
> >         # step 2: make("wood pickaxe") # craft a wood pickaxe using wood from the inventory
> >         ...
> >         Reflecter: Since the goal is to place a table, and placing a table consumes 2 diamonds from the player's inventory according to the game mechanics, the first step is to ensure we have the necessary resources to craft a table, which in this case, are diamonds.
> >         RePlanner: The plan to make a wood pickaxe  is below:
> >         # step 1: mine('grass') # # mine the grass block to potentially get more wood
> >         # step 2: place("table") # place the crafting table on the ground
> >         # step 3: make("wood pickaxe") # craft a wood pickaxe using wood from the inventory
> >         ```
> >
> >     * **Case 4:** The LLM agent continues to search for a tree, when in fact, the nearby stone can be mined without tools. The LLMs need to fully utilize the world-specific rules and the current situation to make optimal decisions, rather than use prior experience to perform inefficient exploration.
> >         ```
> >         You are on the grass.
> >         You see: (object with coordinate)
> >         grass is in front of you.
> >         <grass(-1, 0), grass(1, 0), sand(0, -1), grass(0, 1), lava(-1, -1), water(1, -1), grass(-1, 1), sand(1, 1), stone(4, -1), diamond(4, -2)>
> >         Your status: <health: 6/9, food: 9/9, drink: 8/9, energy: 7/9>
> >         Your inventory: <sapling: 1, coal: 2, diamond: 1>
> >         THINK: The player has been moving around the same area without finding new resources or a tree... Given the game mechanics, it's essential to explore new areas to find a tree for wood, which is crucial for crafting tools and advancing in the game. The player's inventory contains a sapling..., but without wood, the options for crafting are limited. The current strategy should involve moving in directions that haven't been explored yet or where the player hasn't been in a while, aiming to uncover different terrains or resources...
> >         ```

---

> > > ### Author Rebuttal · Authors · 2024-08-16
> > >
> > > > W3: Moreover, I noticed that some implementation details and data statistics are missing...
> > >
> > > We appreciate your feedback regarding the implementation details and data statistics.
> > >
> > > **1. implementation details:**
> > > The detailed hyperparameters of the RL-based baselines are provided in Section 3.2, Lines 187-192. Specifically, for the PPO experiment, we utilize the default parameters from the stable_baselines3 library, while for DreamerV3, we adopt the default parameters as specified in the source code (available at https://github.com/NM512/dreamerv3-torch). All agents are trained for 1 million environment steps with rewards and tested over 20 independent trials. For further details on the LLM baselines, please refer to Lines 193-201. Additional implementation details and prompts can be found in Appendix A and B. We also provide the code including all baselines, at https://github.com/XiaojuanTang/Mars, which can help with reproducing our results.
> > >
> > > **2. data statistics:**
> > > We include comprehensive details in Appendix A of the supplementary material, Mars_NeurIP24.pdf, including the documentation link, environment descriptor of converting visual signals into text, modified commonsense elements and etc. Each frame typically generates a text description with a token length ranging from 120-180 tokens. To ensure the LLM's context window is not overwhelmed, we save the historical trajectory within a certain token limit. Additionally, all experimental world configurations are detailed in Appendix I. Our code repository also includes a demo video for each world to enhance understanding of these configurations.

---

> > > > ### Author Response · Authors · 2024-08-22
> > > > **Looking forward to your reply**
> > > >
> > > > Dear Reviewer V8Vv,
> > > >
> > > > Thank you for your thorough review and insightful comments on our paper. We appreciate your recognition of the motivation of our work in advancing situated inductive reasoning and your suggestions for improvement. We have posted point-to-point replies to each question raised by you, including:
> > > > 1. clarifying the design and reasoning behind the counter-commonsense world modifications in our benchmark;
> > > > 2. conducting manual checks on sampled cases to justify the underlying reasons for the challenges observed in our benchmark;
> > > > 3. providing additional implementation details and data statistics, including hyperparameters used, and a link to our code repository for reproducibility.
> > > >
> > > > Thank you again for your comments and suggestions to improve our paper! We hope that our responses have addressed your concerns. If you have any further questions or comments, please do not hesitate to reach out to us. We value your feedback and look forward to your reply.
> > > >
> > > > Best regards,
> > > >
> > > > Submission 1225 Authors

---

### Official Review · Reviewer_A3jc · 2024-07-24

**Rating:** 7
**Confidence:** 4
**Correctness:** Yes
**Clarity:** Yes

**Review:**

This paper is a well-written paper presents a novel interactive environment called Mars, designed to benchmark and enhance models’ capabilities in situated inductive reasoning. Here are brief reviews of this paper:

Quality: The quality of the work is high, demonstrated through the rigorous design and implementation of the Mars environment. The paper provides a comprehensive evaluation of various RL and LLM-based methods, showcasing the strengths and weaknesses of these approaches in the context of situated inductive reasoning. The introduction of the Induction from Reflection (IfR) method adds depth to the study, highlighting innovative approaches to enhance inductive reasoning.

Clarity: This paper is well-written. The authors effectively communicate the motivation behind the creation of Mars and the specific modifications introduced to the Crafter environment. The structure of the paper is logical, with clear sections that guide the reader through the introduction, methodology, experiments, results, and conclusions. The use of diagrams and tables aids in the understanding of complex concepts and experimental results.

Originality: The originality of the work is significant. Mars is a novel environment specifically designed to challenge and benchmark situated inductive reasoning, a relatively unexplored area in AI research. The counter-commonsense mechanisms introduced in Mars are unique and provide a new dimension to the evaluation of AI models. The Induction from Reflection method is another original contribution, offering a fresh approach to improving inductive reasoning capabilities in AI.

Significance: The significance of this work is substantial. Situated inductive reasoning is crucial for the development of adaptive and context-sensitive AI systems, and Mars provides a valuable testbed for advancing research in this area. The findings from the experiments highlight the limitations of current models and the need for further research and innovation. The paper sets the stage for future studies and potential breakthroughs in inductive reasoning and adaptive AI.

Pros and Cons

Pros:
1. Novel Environment: Mars is a unique and challenging environment for situated inductive reasoning, pushing the boundaries of current AI capabilities.
2. Comprehensive Evaluation: The paper provides a thorough evaluation of state-of-the-art RL and LLM-based methods, offering valuable insights into their performance and limitations.
3.	Innovative Method: The introduction of the Induction from Reflection (IfR) method adds originality and depth to the study, showcasing innovative approaches to enhancing inductive reasoning.
4.	Clear Communication: The paper is well-structured and clearly written, with effective use of diagrams and tables to aid understanding.

Cons:

1.	Performance Limitations: Despite the innovative approach, the overall performance of the models, including IfR, remains suboptimal in the complex Mars environment.
2.	Exploration Efficiency: The limited exploration time and inefficient exploration process could hinder the full potential of the models, suggesting the need for further improvements.
3. Limited changes of the original environment (see the limitation section).

**Strengths:**

Same as pros in the review.

**Additional Feedback:**

Please the the comments above.

**Documentation:**

Yes

**Limitations:**

- The changes of the environment are limited: In the paper, the authors introduce three types of commonsense changes (terrain, survival settings and task dependency) of the crafter environment to become Mars. However, these modifications are limited and not fundamentally change the game mechanisms. As mentioned in line 135-135, authors said "The new world does not introduce additional resources or objects; it only modifies the functions or effects of existing game objects and materials." This makes the contribution of this paper on creating a new environment/benchmark for the field become limited.
- For the experiments using LLMs, the authors only use GPT-4-0125-preview as their LLM. It would be interesting to see how other large language models' performance on this task, especially the open sources models, such as LLaMA, Mistral, Falcon, etc.

**Opportunities For Improvement:**

Major Improvements:
- In the experiments, authors compare their methods with reinforcement learning (RL) and large language model (LLM) based methods. However, there are some emergent works such as https://proceedings.mlr.press/v202/du23f.html and https://arxiv.org/pdf/2303.00001 demonstrated the effectiveness of using LLM for reward design. It would be better for authors to include some of these types of method in their paper as baselines.
- Since Mars uses counter commonsense causalities which is different from the regular knowledge base of "earth", it would be interesting to see using examples to do in-context learning for the LLM first rather than use the LLM directly. If possible, it would be also interesting to see finetune a language model (even not need to be a large language model) with the examples and then run the experiments with the finetuned model since the input to the LLM is just pure text extracted by the wrapper function from the gameplay screen.

Minor Improvements:
- The name of "Mars" is a little bit confusing and misleading. By reading through the whole paper, there is no clear explanation about why it is called "Mars". The only explanation is in line 51-56, which demonstrate that the Mars environment would have different properties compared with the "earth" knowledge (the counter commonsense part), by setting the task dependencies. However, the actual environment is not like Mars which would mislead the readers. For example, Mars would not have the same resources, such as coals or trees. Authors should be careful with the names they are using and prevent confusion of readers.
- The example in the abstract "When you travel to UK, you might initially find ... the new rule." can be removed. It is a great example, but seems redundant in the abstract. Abstract is a concise summary that provides an overview of the main points of the research. The example is unnecessary and it would be better to be included in the introduction.

**Relation To Prior Work:**

Yes

**Summary And Contributions:**

This paper presents a novel interactive environment called Mars, designed to benchmark and enhance models’ capabilities in situated inductive reasoning. Here is a brief summary of the submission and its contributions:
- Introduction of Mars Environment:
     - Mars is built on the foundation of Crafter, an open-world survival game. It introduces various modifications to terrain, survival settings, and task dependencies to create a challenging environment for agents.
     - The modifications adhere to strict principles to ensure resource balance, task achievability, and the utilization of prior knowledge, making the environment stable and playable.
- Situated Inductive Reasoning:
     - The primary focus is on inductive reasoning, where agents need to actively interact with their surroundings, derive useful rules, and perform decision-making tasks in specific contexts.
     - Mars involves counter-commonsense game mechanisms that require agents to adapt to new rules and environments dynamically.
- Evaluation Metrics and Experiments:
     - The paper evaluates various reinforcement learning (RL) and large language model (LLM) based methods, including a novel method called Induction from Reflection (IfR).
     - Experiments show that current models struggle with the challenging situated inductive reasoning tasks in Mars, highlighting the need for improved methodologies.
- Comparative Analysis:
     - Mars is compared with existing benchmarks, emphasizing its unique focus on interactive, situated, and inductive reasoning tasks.
     - The environment’s complexity is demonstrated through modifications in terrain distribution, survival dynamics, and task dependencies, providing a comprehensive testbed for evaluating reasoning abilities.
- Proposed Induction from Reflection Method:
     - IfR forces LLMs to engage in reflective thinking to induce possible game mechanisms based on historical trajectories.
     - The method shows improved performance over other LLM-based methods, underscoring the importance of inductive reasoning in complex, counter-commonsense environments.

Contributions:
- Novel Benchmarking Environment: Mars provides a unique and challenging platform for evaluating and improving models’ capabilities in situated inductive reasoning, which is crucial for adaptive and context-sensitive AI systems.
- Evaluation of State-of-the-Art Methods:
The paper conducts extensive experiments with various RL and LLM-based methods, providing valuable insights into their performance and limitations in the Mars environment.
- Introduction of IfR:
The Induction from Reflection method demonstrates the potential of reflective thinking in enhancing inductive reasoning capabilities, setting a new direction for future research in this domain.
- Comprehensive Analysis:
The comparative analysis with existing benchmarks and the detailed examination of different modifications in Mars contribute to a deeper understanding of the challenges and requirements for effective situated inductive reasoning.

---

> ### Author Rebuttal · Authors · 2024-08-16
>
> Thank you for your insightful and helpful review! We are appreciate your recognition of our contribution in developing a benchmark for situated inductive reasoning. We address your concerns below.
>
> > W1: lack some baselines about using LLM for reward design such as https://proceedings.mlr.press/v202/du23f.html and https://arxiv.org/pdf/2303.00001.
>
> Thank you for helpful comments. We will add the suggested baselines that use LLMs for reward design[1][2]. Consider the time limitation in rebuttal period, we first conduct the experiment of ELLM[1] across "Default", "Survivial", "Task. Dep" and "All three" worlds. To ensure consistency with our experimental setup, we included both intrinsic rewards and health rewards during training. So far, we have run 15,000 episodes and will continue until we have complete results, which we will include in the final version of our paper.
>
> | Mod. Type   |      ELLM    |
> |:----------- |:-------------:|
> | Default     | 5.5 $\pm$ 1.3 |
> | Survival    | 4.9 $\pm$ 1.6|
> | Task. Dep   |  2.9 $\pm$ 0.7|
> | All three.  | 0.9 $\pm$ 1.1 |
>
> The performance of ELLM drops in the Mars compared to the Crafter (Default) environment. We observe that the extent of decline varies with the modification type, with the most substantial drop in the "All three." modification, followed by "Task. Dep" and "Survival". This pattern aligns with the trends we observe using both RL-based and LLM-based methods. These results suggest that while LLM priors can guide RL exploration, when transferring to a novel world with different game mechanics and knowledge, LLMs struggle due to their lack of situated inductive reasoning. This further validates the difficulty of our Mars benchmark under current methods, underscoring the need for more advanced AI systems that can adapt and reason contextually in novel environments.
>
>
> [1] Du, Yuqing, et al. "Guiding pretraining in reinforcement learning with large language models."
>
> [2] Kwon, Minae, et al. "Reward design with language models."
>
>
> > W2: ...it would be interesting to see using examples to do in-context learning for the LLM first rather than use the LLM directly. If possible, it would be also interesting to see finetune a language model (even not need to be a large language model) with the examples and then run the experiments with the finetuned model...
>
> Thanks for your valuable suggestion! In fact, we use few-shot induction examples to prompt LLM for inductive reasoning, as detailed in Appendix I Prompt. Here are the few-shot examples we provided:
> ```
> Reasoning: The player's health decreased by 2 after shot by arrow, indicating that the arrow of skeleton is harmful to the player.
> Mechanism: The arrow of skeleton can cause damage to the player.
>
> Reasoning: The player is facing the water block and cannot enter the water block, indicating that the player cannot swim or the water block is not walkable.
> Mechanism: the water block is not walkable.
>
> Reasoning: The player has been mining the stone block for a long time but has not yet obtained the stone, indicating that the stone block cannot be mined by hand.
> Mechanism: The stone block cannot be mined by hand.
>
> Reasoning: The player has been placing the table in the front stone block for a long time but has not yet placed the table, indicating that the table cannot be placed on the stone block.
> Mechanism: The table cannot be placed on the stone block.
>
> Reasoning: The player can place the table in the front grass block, indicating that the table can be placed on the grass block.
> Mechanism: The table can be placed on the grass block.
> ```
> Despite providing these few-shot induction demonstrations, LLMs still perform poorly in inducing new rules in novel scenarios. This is likely because in-context learning is heavily dependent on the similarity of provided examples to the target task and the coherence of data distribution[1][2][3][4]. When LLMs are required to induce rules that are distinct from the examples in a novel scenario, it becomes difficult for them to perform inductive reasoning effectively through in-context learning.
>
> Regarding your suggestion to fine-tune LM with examples from different worlds to help it master inductive reasoning, we think this is an excellent idea. However, fine-tuning requires elaborately constructing the appropriate dataset, and due to time constraints during the rebuttal period, we will leave this as future work. We greatly appreciate your suggestion.
>
>
> [1] Min, Sewon, Xinxi Lyu, Ari Holtzman, Mikel Artetxe, Mike Lewis, Hannaneh Hajishirzi, and Luke Zettlemoyer. "Rethinking the role of demonstrations: What makes in-context learning work?."
>
> [2] Jiachang Liu, Dinghan Shen, Yizhe Zhang, Bill Dolan, Lawrence Carin, and Weizhu Chen. "What makes good in-context examples for gpt-3?"
>
> [3] Zihao Zhao, Eric Wallace, Shi Feng, Dan Klein, and Sameer Singh. "Calibrate before use: Improving few-shot performance of language models."
>
> [4] Ekin Akyürek, Dale Schuurmans, Jacob Andreas, Tengyu Ma, and Denny Zhou. "What learning algorithm is in-context learning? investigations with linear models."
>
> > W3: The name of "Mars" is a little bit confusing and misleading. By reading through the whole paper, there is no clear explanation about why it is called "Mars".
>
> We apologize for any confusion caused by the name "Mars". To clarify, "Mars" in our paper is not meant to represent the actual planet Mars. Instead, we use "Mars" symbolically to represent an environment with knowledge and conditions that differ from commonsense (or "Earth" knowledge), such as unconventional terrain and survival settings.

---

> > ### Author Rebuttal · Authors · 2024-08-16
> >
> > > W4: The example in the abstract "When you travel to UK, you might initially find ... the new rule." can be removed. It is a great example, but seems redundant in the abstract. Abstract is a concise summary that provides an overview of the main points of the research. The example is unnecessary and it would be better to be included in the introduction.
> >
> > Thank you for your helpful suggestion. We included this example in the abstract to help readers quickly grasp the concept of situated inductive reasoning. We believe it provides immediate clarity, making the abstract more accessible. However, we will also make further revisions to ensure the abstract remains concise and focused.
> >
> >
> > > W5: The changes of the environment are limited. The new world does not introduce additional resources or objects; it only modifies the functions or effects of existing game object
> >
> > Thank you for your insightful comment. Yes, our modifications to the Crafter environment do not introduce additional new resources or objects. This is because our primary focus is on evaluating models' situated inductive reasoning abilities in an adaptive and context-sensitive manner, which need agents quickly derive new general rules from current situation and apply the newly acquired knowledge effectively. By not fundamentally changing the game mechanisms, we ensure that agents have the opportunity to adapt more quickly to the new environment rather than having to learn new concepts or objects from scratch. However, we acknowledge that this may lead to a lack of diversity in the modifications. In the future, we plan to incorporate more diverse elements to enhance the richness of this environment.
> >
> > > W6: For the experiments using LLMs, the authors only use GPT-4-0125-preview as their LLM. It would be interesting to see how other large language models' performance on this task, especially the open sources models, such as LLaMA, Mistral, Falcon, etc.
> >
> > Thank for your helpful suggestions. We conduct additional experiments with the open-source model LLaMA-3.1-8B-instruct. We evaluate both the ReAct and IfR models across different worlds, using the same prompts and hyperparameters as with GPT-4. The results show that LLaMA's performance declines when encountering the Mars environment. Additionally, our model IfR consistently outperforms ReAct across all scenarios. These findings align with the results obtained using GPT-4, further validating the importance of inductive reasoning and highlighting the challenges posed by our benchmark.
> >
> >
> > | Mod. Type   |     ReAct     |  Ours (IfR)   |
> > |:----------- |:-------------:|:-------------:|
> > | Default     | 3.6 $\pm$ 2.1 | 3.8 $\pm$ 2.4 |
> > | Terrain     | 2.1 $\pm$ 2.2 | 3.8 $\pm$ 2.1 |
> > | Survival    | 2.3 $\pm$ 2.5 | 3.7 $\pm$ 2.8 |
> > | Task. Dep   | 2.3 $\pm$ 1.0 | 2.9 $\pm$ 1.0 |
> > | Terr. Surv. | 1.1 $\pm$ 1.4 | 3.8 $\pm$ 2.0 |
> > | Terr. Task. | 3.0 $\pm$ 1.6 | 3.3 $\pm$ 1.2 |
> > | Surv. Task. | 0.7 $\pm$ 2.0 | 1.1 $\pm$ 1.3 |
> > | All three.  | 0.2 $\pm$ 1.2 | 0.8 $\pm$ 1.4 |

---

> > > ### Author Response · Authors · 2024-08-22
> > > **Looking forward to your reply**
> > >
> > > Dear Reviewer A3jc,
> > >
> > > Thank you for your positive and constructive comments, which are very helpful and make our paper stronger. We have posted point-to-point replies to each question raised by you, including:
> > >
> > > 1. adding the suggested baselines that use LLMs for reward design;
> > > 2. providing additional experiments with the open-source model LLaMA-3.1-8B-instruct;
> > > 3. addressing the use of few-shot demonstrations for inductive reasoning, and we provide the examples used in our experiments;
> > > 4. other minor questions
> > >
> > > Thank you again for your comments and suggestions to improve our paper! We hope that our responses have addressed your concerns. If you have any further questions or comments, please do not hesitate to reach out to us. We value your feedback and look forward to your reply.
> > >
> > > Best regards,
> > >
> > > Submission 1225 authors

---

> > ### Comment · Reviewer_A3jc · 2024-08-27
> >
> > Thanks for all the detailed explanation and additional experiments for the rebuttal from authors. I do not have further questions and I will keep my score.

---

> > > ### Author Response · Authors · 2024-08-27
> > > **Response to Reviewer A3jc**
> > >
> > > Thank you for your updated feedback. We are pleased that our rebuttal and the additional experiments address your concerns. If you have any further questions, please feel free to reach out.

---

### Official Review · Reviewer_56w9 · 2024-08-01
**Mars: Situated Inductive Reasoning in an Open-World Environment**

**Rating:** 7
**Confidence:** 3
**Correctness:** No overt errors outside of grammar an…

**Review:**

The Mars game environment provides a place to undermine and subvert the rules of the Crafter game in ways which allows exploring the heuristics agents use to play the game. Overall this feels like an indirect an inefficient manner in which to test LLM agents ability to observe and construct an internally consistent set of rules about the game universe. Specifically, the LLM agents start with minimal foundation knowledge about the game and must effectively button mash until something happens. At which point the rule libraries come into play to try and record what worked in the past. A more direct measure of this might be providing the text transcript of N game interactions taken be a human to the LLM agent, as a primer of what is possible; and then allow it to use the rule library it built up (or any of the other prompting methods described in the paper) to take a turn at playing that specific instance of Mars modifications from the start to understand how well it was able to figure out from the shown examples what the situated reasoning is. To borrow the metaphor from the abstract, when traveling to the UK and driving, you don't start out with minimal knowledge about how to operate a car and minimal understanding of the rules of the road. You already know how to play a similar game, and you just need to adapt. Without fine-tuning the LLMs (or otherwise setting up a baseline agent which is good at the base Crafter game); the evaluation isn't exploring the situated inductive reasoning alone, it has to also content with the system noise inherit to operating in a game the agent is unfamiliar with.

**Strengths:**

This paper presents a methodology for varying the difficulty and analyzing the success of LLM agents operating in the game world, where the LLMs need to be able to ingest a summarized history of actions taken and understand the rules which govern the world of the game. The prompting strategies described build out a framework of rules and heuristics that the LLMs can leverage as a crutch to operate successfully in an unfamiliar game environment.

**Additional Feedback:**

N/A

**Clarity:**

The overall presentation of the ideas in the paper need improvement for clarity. The narrative being told and the presentation of the work impeeds the readers understanding. Some of the sections are clearly written (for example the related works). For other sections its harder to follow the narrative thread you are presenting about the work.

**Documentation:**

Acceptable. The Github link to the code is provided, and appendix documents the changes to Crafter game, as well as the LLM prompting strategies.

**Limitations:**

The inclusion of RL based methods is puzzling, as they are directly trained on the modified Mars env. Therefore they provide no insight into model ability to perform situated inductive reasoning. The RL methods only provide a baseline of what is learning able for a model explicitly trained on the modified environment. Additionally, the PPO based methods perform very poorly (per table 2), and the DreamerV3 methods are approximately the same quality as some of the LLM based methods.  To me these RL results indicate the upper bound of performance which might be expected of any given Mars environment (i.e. the env difficulty).

The biggest limitation of this work is it doesn't clearly evaluate LLM agents ability to perform situated inductive reasoning. The LLM agents used in the experiments are relying heavily on the LLM models base understanding of Minecraft to frame the basic game mechanics. From there the history and state of the game shown to the agents allows the extraction of heuristics about the operation of the game world. Whilst text about Minecraft likely is significantly present in the training data for the LLMs, the liklihood that those models have direct experience with the Crafter game environment is much lower. So the fundamental thing being evaluated is how well does the Minecraft knowledge transfer to Crafter. A more direct experimental setup focused solely on situated inductive reasoning as the game rules change, would be to fine tune an LLM agent to play Crafter well, and then see how it does under the varying prompting strategies to adapt the learned game strategies to the modified Mars world.

For example, Line 54: "However, they cannot merely leverage their prior knowledge (such as “consuming cows increases health”) since these pre-stored “earth” knowledge might no longer apply on Mars." The LLM relying on Minecraft rules in the training dataset has a longer chain of operations to conclude cows are related to health gain, you need to kill the cow, collect the meat, build an oven, and cook the meat before it becomes useful.

**Opportunities For Improvement:**

Correct the word usage for "induce" which is used several times as a shorthand for the process of induction based on observations, which is not what the word "induce" means. For example lines 56, 80, and Figure 1 caption.

Typo line 68 "relexion" -> "reflexion"

**Relation To Prior Work:**

The related works provides an acceptable overview.

**Summary And Contributions:**

This paper proposes a 2D Minecraft like open world game as a method for measuring the ability of models to figure out and leverage the foundational rules that govern the game world, i.e. perform situated inductive reasoning. The authors describe the modifications from the base game Crafter, how those modifications are constrained to make the game winnable, and how modifications cause assumptions the model might make about game mechanics to be incorrect. The authors then benchmark 2 RL based methods (trained on the exact modified environment) as well as 4 LLM-based prompting strategies to formulate the game into a text-based interface the LLM can operate with (including a novel strategy by the authors).

---

> ### Author Rebuttal · Authors · 2024-08-16
>
> Thank for your careful review of our paper and insightful questions. We will address your concerns below.
>
> > W1: The biggest limitation of this work is it doesn't clearly evaluate LLM agents ability to perform situated inductive reasoning... Whilst text about Minecraft likely is significantly present in the training data for the LLMs, the liklihood that those models have direct experience with the Crafter game environment is much lower... Since the models are not good at playing Crafter well, the evaluation may not isolate situated inductive reasoning from the noise introduced by the agents’ unfamiliarity with the game.
>
> Thank you for your insightful feedback regarding the evaluation of LLM agents' ability to perform situated inductive reasoning. LLMs are pre-trained on vast and diverse textual data, which provides them with extensive world knowledge and commonsense information. This knowledge often aligns with the mechanisms of the Crafter game, which is why **many studies leverage the commonsense knowledge encoded in LLMs to guide RL for more efficient exploration in Crafter**. For instance, ELLM[1] shapes rewards towards commonsense and useful behaviors through a pretrained LLM, while AdaRefiner[2] uses sub-goals suggested by the LLM to guide exploration.
>
> To further validate the LLMs' understanding of Crafter's game mechanics, we conduct two additional experiments:
>
> * 1. Knowledge Mastery Quiz:
> To assess whether LLMs have internalized Crafter's knowledge, we design a [quiz](https://anonymous.4open.science/r/Mars-6418/quiz.json) consisting of 72 multiple-choice questions on Crafter's world knowledge. **GPT-4 achieves an 81\% accuracy rate**, indicating that LLMs encode a significant portion of Crafter's game knowledge.
>         ```
>         Prompt:
>         I will give you a multiple-choice question to test your commonsense knowledge. Please choose the correct answer from the options. If you do not know the answer, please output "I don't know". The response format is below:
>         Reasoning: {your reasoning}
>         Answer: {your answer}
>         ```
> * 2. In-Context Knowledge Experiment:
> We also experiment with the ReAct model by providing it with Crafter's game knowledge in-context. We observe **minimal performance improvement** with this knowledge compared to without it. Performance dropped further when transitioning from the Default to Mars scenarios, highlighting the challenges of adapting to novel situations. Interestingly, when modifying "Task. Dep" type, providing knowledge led to poorer performance, which may be due to the emphasis on in-context commonsense knowledge making it more difficult to process counter-commonsense situations, further disrupting its ability to perform situated inductive reasoning.
>         | Mod. Type   | with in-context knowledge | without in-context knowledge |
>         |:----------- |:-------------------------:|:----------------------------:|
>         | Default     |       7.9 $\pm$ 2.7       |        7.7 $\pm$ 1.6         |
>         | Terrain     |       7.8 $\pm$ 3.1       |        7.4 $\pm$ 2.7         |
>         | Survival    |      7.0 $\pm$ 4.1       |        6.4 $\pm$ 3.7         |
>         | Task. Dep   |       1.8 $\pm$ 0.5       |        5.0 $\pm$ 2.1         |
>         | Terr. Surv. |       6.8 $\pm$ 1.7       |        6.7 $\pm$ 2.5         |
>         | Terr. Task. |       4.4 $\pm$ 0.9       |        4.8 $\pm$ 2.0         |
>         | Surv. Task. |       0.8 $\pm$ 0.5       |        1.5 $\pm$ 1.3         |
>         | All three.  |       0.1 $\pm$ 0.8       |        0.7 $\pm$ 1.6         |
>
>
>
>
>
> Furthermore, situated inductive reasoning not only emphasizes *adaptability* across different situations but also *situatedness*, which involves summarizing and forming conclusions from current and live observations (Line 31). In other words, the agent needs to **dynamically understand the current situation** and quickly derive new general knowledge (rules) that can be **applied effectively in a new context**. Therefore, even if the model does not fully grasp the Crafter environment, it can still utilize situated inductive reasoning ability to master the remaining aspect.
>
> In conclusion, we emphasize that the stored knowledge in LLMs often aligns with Crafter's game mechanics. To support this, we reference studies that leverage LLMs' commonsense knowledge to guide RL exploration effectively in Crafter. Additionally, we design a quiz to evaluate whether LLMs have internalized Crafter's knowledge. Furthermore, we conduct experiments where Crafter-specific knowledge was provided in context, revealing minimal performance improvement and a decline in performance across novel scenarios. Lastly, in addition to adaptiveness, we underscore the importance of situatedness in situated inductive reasoning, where agents need dynamically understand and apply new knowledge in varied contexts.
>
>
>
>
> quiz link: [https://anonymous.4open.science/r/Mars-6418/quiz.json](https://anonymous.4open.science/r/Mars-6418/quiz.json)
>
> [1] Du, Yuqing, et al. "Guiding pretraining in reinforcement learning with large language models."
>
> [2] Zhang, Wanpeng, and Zongqing Lu. "AdaRefiner: Refining Decisions of Language Models with Adaptive Feedback."

---

> > ### Author Rebuttal · Authors · 2024-08-16
> >
> > > W2: The inclusion of RL based methods is puzzling, as they are directly trained on the modified Mars env.
> >
> > We appreciate your observation and agree with your point.
> >
> > 1. As mentioned in Lines 180-183, "Note that RL-based methods individually train a model for each world with 1 million training steps. They do not truly solve the problem of quickly adapting to new environments in situated inductive reasoning scenarios. These experiments were conducted solely for reference purposes."
> > 2. Furthemore, to assess the situated inductive reasoning capabilities of RL-based methods, we also evaluate the modified worlds using DreamerV3 trained in the Crafter environment (as detailed in Appendix D). Our findings show that while it performs well in Crafter, it struggles to quickly adapt to counter-commonsense worlds, even when the differences are minor. These results further illustrate that RL-based methods do not adequately address the challenges of situated inductive reasoning.
> >
> > > W3: Correct the word usage for "induce" which is used several times as a shorthand for the process of induction based on observations, which is not what the word "induce" means. For example lines 56, 80, and Figure 1 caption.
> >
> > Thank you for your feedback regarding the usage of the word "induce". We would like to clarify that the term "induce" is used correctly in our paper. As per the definition provided by the Collins English Dictionary (https://www.collinsdictionary.com/dictionary/english/induce), "induce" can mean "to assert or establish (a proposition about a class of phenomena) on the basis of observations on a number of particular facts", which aligns with the inductive reasoning process described in our work. We believe this usage is appropriate and precise for our context. Hope this clarification addresses your concerns.
> >
> >
> > > W4: Typo line 68 "relexion" -> "reflexion"
> >
> > Thanks for pointing out the typo. We promise to correct it.
> >
> > > W5: The overall presentation of the ideas in the paper need improvement for clarity. The narrative being told and the presentation of the work impeeds the readers understanding. Some of the sections are clearly written (for example the related works). For other sections its harder to follow the narrative thread you are presenting about the work.
> >
> > Thank you for your feedback on the clarity and presentation of our paper. We will take your comments into careful consideration and further refine the paper to enhance readability and ensure a clearer narrative thread throughout.

---

> > > ### Comment · Reviewer_56w9 · 2024-08-16
> > >
> > > The Knowledge Mastery Quiz baseline quiz exploring how much the modified Crafter env is present in the commonsense knowledge of the LLM is a welcome improvement, and helps bring the situated inductive reasoning arguments into focus.
> > >
> > > For the In-Context Knowledge experiment its interesting that there is a minor performance bump (with higher variance) for most Mod Types, with the exception of task dependency. That is beginning the exploration of what type of reasoning and modifications the tested models are most robust (or weak) against. Combined with the performance on Terr. Surv. the included table indicates the action order relationships is by far the weakest component of LMs.
> > >
> > > It would be great to include the results you showed on the "Knowledge Mastery Quiz" and the "In-Context Knowledge Experiment" in the manuscript.
> > >
> > > Both of these additions address my major concerns about focusing the manuscript on the stated objective of exploring situated inductive reasoning. Combined with the responses to the other reviewers, I have raised my review score from 5 to 7.

---

> > > > ### Author Response · Authors · 2024-08-16
> > > >
> > > > Thanks for your positive feedback! We are delighted to hear that our rebuttal and additional experiments are helpful, and we will include them in our revision. We sincerely appreciate your decision to raise the score.

---

### Author Rebuttal · Authors · 2024-08-16

## General Response

We sincerely thank the reviewers for their thoughtful feedback and for recognizing the potential of our research in advancing situated inductive reasoning. In response the reviewers' comments and suggestions, we conduct additional experiments and make several improvements to our paper:
1. we design a quiz to evaluate whether LLMs have internalized Crafter's knowledge. GPT-4 achieves 81\% accurate rate. (Reviewer 56w9)
2. we conduct experiments where Crafter-specific knowledge is provided in context, revealing minimal performance improvement and a decline in performance across novel scenarios. (Reviewer 56w9)
3. we add the suggested baselines that use LLMs for reward design[1]. (Reviewer A3jc)
4. we conduct additional experiments with the open-source model LLaMA-3.1-8B-instruct to evaluate both the ReAct and IfR models across different worlds. (Reviewer A3jc)
5. we further elaborate on the challenges presented by our Mars benchmark through detailed case studies. (Reviewer V8Vv)

Once again, we thank the reviewers for their valuable input, which has significantly contributed to the improvement of our paper. We believe these additions and clarifications strengthen our work and address the concerns raised. We look forward to your further feedback.

[1] Du, Yuqing, et al. "Guiding pretraining in reinforcement learning with large language models."

---

### Decision · Program_Chairs · 2024-09-26

**Decision:**

Accept (Poster)

**Comment:**

This paper proposes a novel open world game as a method for measuring the ability of models to figure out and leverage the foundational rules that govern a game world after training on a similar but distinct environment (that is, it introduces ‘Martian’ modifications such as terrains and tasks to the simulation game Crafter). The agent must perform inductive reasoning in which they derive rules from environment interaction and perform tasks in the novel context. This testbed is then empirically evaluated for utility using two reinforcement learning-based agents and four LLM-based agents, including one novel approach (‘induction from reflection’). Reviewers generally agree that the work is well written and well motivated, that the demonstrations of agent performance in the environment are rigorous, and that the novel approach proposed is of interest. However, the environmental changes proposed may be too distinct from common sense behavior; a benchmark in which two real but different sets of rules apply would be more robust and less in need of evaluation as to the source of performance failures. Limitations to the differences of the novel environment are also a source of concern, although this is somewhat alleviated by acknowledgement of these limitations.

Strengths:
- The proposed testbed is indeed novel, and addresses a problem space (situated inductive reasoning) that is relevant to a variety of real-world agent performance problems. The work is original and explores a relatively underexplored problem space.
- The innovative induction from reflection approach represents an original contribution in its own right, and its performance suggests potential value in other problem spaces.
- The paper is well written and clear, and does a good job of motivating the work performed and explaining the approaches tried.

Weaknesses:
- The overall performance of the models explored remain suboptimal. This work does not claim to solve situated inductive reasoning.
- The changes to the world are not enormous, limiting the scope of the reasoning and learning required.
- The environmental changes proposed may be too distinct from common sense behavior; a benchmark in which two real but different sets of rules apply would be more robust.
- Additional manual evaluation of failure cases to determine which of several possible sources of failure attain would be informative.